# Quadratic Magnetic Gradients from 7- and 9-Spacecraft Constellations

Chao Shen[1], Gang Zeng[2], Rungployphan Kieokaew[3], Yufei Zhou[1]

[1]School of Science, Harbin Institute of Technology, Shenzhen 518055, China
[2]School of Mathematics and Physics, Jingchu University of Technology, Jingmen, China
[3]Institut de Recherche en Astrophysique et Planétologie (IRAP), France

*Correspondence to*: Chao Shen (shenchao@hit.edu.cn), Gang Zeng (gzeng2014@126.com)

**Abstract.** To uncover the dynamics of magnetised plasma, it is crucial to determine the geometric structure of the magnetic field, which depends on its linear and quadratic spatial gradients. Estimating the linear magnetic gradient requires at least four simultaneous magnetic measurements, while calculating the quadratic gradients generally requires at least ten. This study focuses on deriving both linear and quadratic spatial gradients of the magnetic field using data from the nine-spacecraft HelioSwarm or seven-spacecraft Plasma Observatory constellations. Time-series magnetic measurements, combined with transformations between reference frames, were employed to determine the apparent velocity of the magnetic structure and the quadratic magnetic gradient components along the direction of motion. The linear gradient and remaining components of the quadratic gradient were derived using the least-squares method, with iterative calculations applied to ensure precision. The validity of the approach was demonstrated using magnetic flux ropes and dipole magnetic field models. The findings indicate that constellations with at least seven spacecraft in nonplanar configurations can successfully yield linear and quadratic spatial gradients of magnetic field.

**Key Points:**

1. An iterative algorithm for the quadratic magnetic gradient based on measurement with constellations comprising at least seven spacecraft is presented.
2. Magnetic flux ropes and dipole magnetic field testing demonstrated the validity of the approach.
3. Constellations containing at least seven spacecraft with nonplanar configurations are required for the approach.

**Key Words:** Multiple Spacecraft Measurements, Space Plasmas, Magnetic field, Quadratic Magnetic Gradient, Least Squares Method

# 1 Introduction

Multi-spacecraft constellations provide a unique capability to observe plasma processes at various spatiotemporal scales simultaneously. In particular, in situ magnetic measurements from such constellations enable the deduction of magnetic gradients, allowing for the investigation of fine magnetic structures, current densities, and magnetic field geometries. Typically, magnetic measurements from at least four spacecraft in a nonplanar configuration are required to deduce the three-dimensional (3-D) linear spatial gradient of a magnetic field (Harvey, 1998; Chanteur, 1998; Chanteur and Harvey, 1998; Shen et al., 2003;

De Keyser, et al., 2007; De Keyser, 2008; Hamrin et al., 2008; Shen et al., 2012). Additionally, linear spatial gradients of other scalar fields (e.g., plasma moments) or vector fields (e.g., an incompressible velocity field) can be obtained similarly. This is done by performing a Taylor expansion around the origin (e.g., the four-spacecraft mesocentre) up to the first order; the linear gradient, which provides a (unique) solution that fits the measurements, is then obtained using the least-squares method (Harvey, 1998; Chanteur, 1998; Chanteur and Harvey, 1998; Shen et al., 2003; Broeren et al., 2021).

The Cluster mission (Escoubet et al., 1997, 2001) and the Magnetospheric MultiScale (MMS; Burch et al., 2015) mission both utilize four-spacecraft constellations arranged in a tetrahedral configuration. Using the simultaneous magnetic measurements from these missions allows the linear spatial gradient of the magnetic field, e.g., the current density distribution, to be estimated and the topology of the magnetic field to be further derived (Dunlop et al., 2002b; Shen et al., 2003, 2008, 2012, 2014; Shi et al., 2005; Runov et al., 2006; Shi et al., 2010; Zhang et al., 2011; Rong et al., 2011; Burch and Phan, 2016;

Dong et al., 2018; Pitout and Bogdanova, 2021; Haaland et al., 2021). Furthermore, four-point magnetic field measurements can also be applied to determine the orientation and motion of planar discontinuities (Russell et al., 1983; Dunlop et al., 2002a; Sonnerup et al., 2004), as well as the geometry of curved boundary layers (Kieokaew et al., 2018; Kieokaew and Foullon, 2019; Shen et al., 2020). For a planar constellation or a constellation comprising three spacecraft, only a two-dimensional linear magnetic gradient in the constellation plane can generally be derived (Vogt et al., 2009, 2013; Shen et al., 2012).

Nevertheless, for certain structures such as one-dimensional and force-free structures, magnetic measurements from planar constellations or even Double Star constellations can also be reduced to a three-dimensional linear magnetic gradient (Vogt et al., 2009, 2013; Shen et al., 2012).

To estimate second spatial derivatives of the magnetic field (or Hessian matrix over each component of the magnetic field), simultaneous magnetic measurements from a constellation with more spacecraft are required. Considering a Taylor's

expansion of the magnetic field around the origin up to the second order, there are 10 unknown parameters: 1 magnetic measurement at the origin, 3 components of linear magnetic gradient, and 6 components of second-order magnetic gradient (i.e., the quadratic gradient tensor is symmetric). To obtain a unique solution to the system of equations, we need a number of unknown parameters to be equal to, or less than, the number of the constraints (i.e., equations). Therefore, 10-point measurements are required to solve the quadratic gradient (Chanteur, 1998; Shen et al., 2021b) given that not all spacecraft

are simultaneously within the same quadratic surface (Zhou and Shen, 2024). Nevertheless, the quadratic gradient of a

magnetic field can still be estimated from four-spacecraft constellations if additional current density measurements deduced from electron and ion measurements and certain physical constraints such as Ampère's law and Magnetic Gauss's law, are utilised (Liu et al., 2019; Torbert et al., 2020; Denton et al., 2020; Shen et al., 2021a). Utilising the linear and quadratic gradients of the magnetic field means that the complete geometry of a magnetic field, which concerns linear, e.g., current sheets, and nonlinear spatial structures, e.g., magnetic flux ropes, can be determined (Shen et al., 2021a). Furthermore, the calculation of quadratic spatial gradients of physical electromagnetic and plasma quantities in general allows us to study nonlinear plasma dynamics involving second-order spatial derivatives such as in plasma turbulence (e.g., Politano and Pouquet, 1998a, b; Yang, 2019; Pecora et al. 2023) and nonlinear wave dynamics (e.g. Chian et al. 1998; 2022), among others.

The HelioSwarm mission (Klein et al. 2023) is a nine-spacecraft constellation consisting of one hub (mothercraft) and eight nodes (daughtercraft) planned to be launched in 2029 by the National Aeronautics and Space Administration (NASA). The swarm of nine spacecraft will allow simultaneous cross-scale observations of turbulent solar-wind plasmas for the first time in the vicinity of Earth. Specifically, each spacecraft of HelioSwarm will be equipped with a Fluxgate magnetometer and a Search-Coil magnetometer, allowing comprehensive measurements of magnetic fields at 9 points simultaneously. Plasma Observatory (Retinò et al. 2022) is a new European Space Agency (ESA) mission with a seven-spacecraft constellation in the Solar-Terrestrial environment, currently under Phase-A study. One important topic for these two new multi-spacecraft constellations is to ascertain how the linear and quadratic gradients of the magnetic field can be inferred from seven- or nine-point magnetic measurements, allowing the fine, nonlinear spatial structures of the magnetic field in a space plasma to be identified. In this study, a new algorithm for calculating the linear and quadratic spatial gradients of the magnetic field from 7- or 9-point simultaneous magnetic measurements was derived using the least-squares method. By considering the transformation of reference frame involving mixed space-time derivatives of the magnetic field, we demonstrate that 7- or 9-point simultaneous measurements can be used to estimate quadratic spatial gradients. Here, by exploiting the least-squares method, we propose an iterative approach to achieve an optimal solution.

The remainder of this paper is as follows. The new algorithm for calculating the linear and quadratic magnetic gradients from 7- or 9-point simultaneous magnetic measurements is presented in Section 2; a description of the tests conducted for two typical nonlinear magnetic structures (a cylindrical force-free flux rope and a dipole magnetic field), which were utilized to check the validity and accuracy of the new algorithm, is given in Section 3; the accuracy of the algorithm is evaluated in section 4; and finally, the conclusions are presented in Section 5.

## 2. Methodology

### 2.1 The scheme

Calculation of the linear and quadratic gradients of a magnetic field generally requires simultaneous magnetic measurements from at least ten spacecraft. There are 3+9+18=30 parameters in the Taylor expansion up to second order, and

3N magnetic field measurements in an array with N spacecraft are needed accordingly. Thus, using the magnetic measurements of nine-spacecraft (9S/C) HelioSwarm or seven-spacecraft (7S/C) Plasma Observatory constellation means that additional constraints are required. The transfer relationships between different reference frames are the proper limitations used for completely determining the spatial linear and quadratic gradients of the magnetic field. In these limits, we assume that the magnetic structures are slowly evolving during their passages through the multi-point constellations such that any differences in the measurements at different spacecraft can be attributed to the spatial variations rather than the temporal changes (i.e., evolution of magnetic structures).

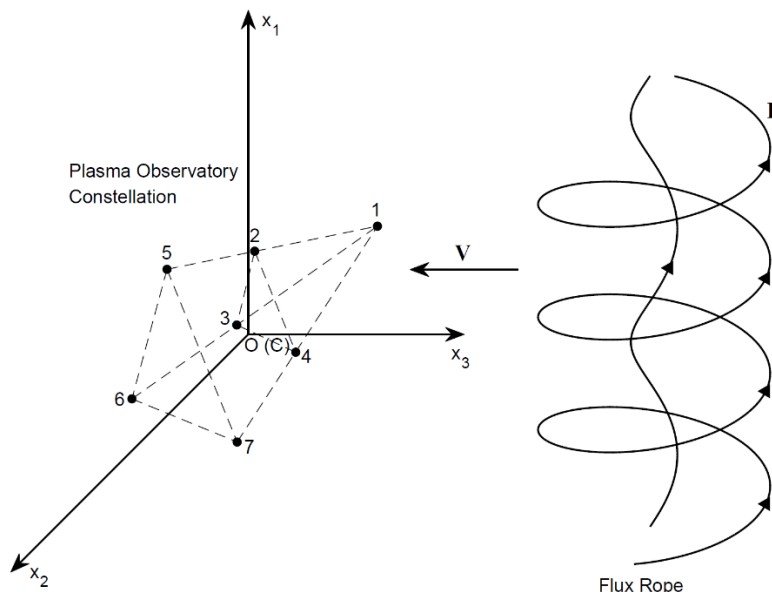

**Figure 1. Schematic plot showing observation of a magnetic structure by the Plasma Observatory Constellation, which is composed of seven spacecraft. (The special configuration is similar to the mission term proposal and the actual geometry can deviate from it.) Barycentric coordinates are adopted; thus, the centre C of the constellation overlaps with the origin O of the Cartesian coordinates $(x_1, x_2, x_3)$ the magnetic structure is assumed to be moving at velocity V relative to the constellation, and the $x_3$ axis is presumed to be anti-parallel to V.**

The Taylor expansion of the magnetic field within two orders is expressed using

$$\mathbf{B}(t,\mathbf{r}) = \mathbf{B}(t,\mathbf{r_c}) + (\mathbf{r}\text{-}\mathbf{r_c}) \cdot \nabla \mathbf{B}(t,\mathbf{r_c}) + \frac{1}{2}(\mathbf{r}\text{-}\mathbf{r_c}) \ (\mathbf{r}\text{-}\mathbf{r_c}) \cdot \nabla\nabla \mathbf{B}(t,\mathbf{r_c}).$$ (1)

The Taylor expansion of each component of the magnetic field at each spacecraft $\alpha$ can be written as

$$f_{(\alpha)} = f_c + x^i_{(\alpha)}(\nabla_i f)_c + \frac{1}{2} x^i_{(\alpha)} x^j_{(\alpha)}(\nabla_i \nabla_j f)_c = f_c + x^i_{(\alpha)} g_i + \frac{1}{2} x^i_{(\alpha)} x^j_{(\alpha)} G_{ij},$$ (2)

where $f$ represents any one of the three components $B^1, B^2, B^3$ of the magnetic field **B**. The first-order gradient is denoted as $g_i \equiv (\nabla_i f)_c$, where $i$ =1, 2, 3, i.e. the three cartesian components, and the second-order gradient is denoted as

$G_{ij} \equiv (\nabla_i \nabla_j f)_c$ where $i, j$=1, 2, 3.

Conventionally, 10-point simultaneous measurements are necessary to infer both the first- and second-order spatial gradients of a physical scalar field (Chanteur, 1998; Shen et al., 2021b). To obtain such the spatial gradients with the 9S/C HelioSwarm and 7S/C Plasma Observatory, we consider adding additional physical constraints to the system of equations. Figure 1 shows a schematic diagram about observation of a magnetic structure by the Plasma Observatory Constellation. The

shape of the constellation is ideal but this does not change the generality and applicability of our method.

The following transformation relationship involving the mixed space-time derivatives is used for the magnetic measurements:

$$\partial_t \mathbf{B} = -\mathbf{V} \cdot \nabla \mathbf{B} \text{ and } \partial_t \nabla \mathbf{B} = -\mathbf{V} \cdot \nabla \nabla \mathbf{B}. \tag{3}$$

By computing the temporal derivative $\partial_t \boldsymbol{B}$, and then the temporal derivative of spatial gradient $\partial_t \nabla \boldsymbol{B}$, this relationship allows

both the apparent velocity **V** of the magnetic structure and the nine components of the quadratic magnetic gradient tensor along the direction of motion, $\hat{\mathbf{V}} \cdot \nabla \nabla \mathbf{B}$, to be obtained (Shen et al., 2021a). The constraints to the Eq. (3) are that the plasmas are highly conductive and have a very low velocity (V/c ≪1, where V is the apparent speed of the magnetic structure and c is the speed of light in vacuum), and the physical processes are slowly evolving at low frequencies. The truncation errors in formula (3) are on the order V/c.


### 2.1.1 The zeroth iteration

First, the temporal variation rate $\partial_t \mathbf{B}$ and first-order magnetic gradient $(\nabla \mathbf{B})^{(0)}$, where the uppercase label (0) denotes the zeroth order, can be obtained from seven- or nine-point simultaneous magnetic measurements. Here, the change rate of the magnetic field can be obtained from the temporal (time-series) measurements at each spacecraft. The linear spatial

gradient can be obtained using four-spacecraft techniques (Chanteur, 1998; Harvey, 1998; Shen et al. 2003). Using equation (3), we can thus obtain the apparent velocity **V** of the magnetic structure (Shen et al., 2021a). Next, the longitudinal components of the second-order magnetic gradient, $\hat{\mathbf{V}} \cdot \nabla \nabla \mathbf{B} = \frac{1}{V} \partial_3 \nabla \mathbf{B}$, can be deduced from the transformation relationship (3). These steps will be described in detail in Section 2.2.1. Finally, the remaining nine components of the second-order magnetic gradient (i.e., the transverse components $G_{rs}^{(1)} = (\nabla_r \nabla_s f)^{(1)}$ where ($r, s$ =1, 2)) can be determined from the seven- or nine-point

simultaneous magnetic measurements using the least-squares method, allowing a first-order quadratic magnetic gradient $(\nabla\nabla\mathbf{B})^{(1)}$ to be obtained, as described next.

### 2.1.2 The first order iteration

Provided with the first-order quadratic magnetic gradient $(\nabla\nabla\mathbf{B})^{(1)}$, the corrected first-order linear magnetic gradient $(\nabla\mathbf{B})^{(1)}$ can be found using the least-squares method. Furthermore, the corrected apparent velocity $\mathbf{V}^{(1)}$ of the magnetic

structure and the longitudinal components of the second-order quadratic magnetic gradient $(\hat{\mathbf{V}}\cdot\nabla\nabla\mathbf{B})^{(2)}$ can be obtained from the transformation relationship (3). Again, the corrected transverse components of the quadratic magnetic gradient $(G_{rs}^{(2)}(r,s=1,2))$ are obtained using the least-squares method, allowing a second-order quadratic magnetic gradient $(\nabla\nabla\mathbf{B})^{(2)}$ to be obtained.

The iterations are performed repeatedly until results are converge, which means satisfactory results are achieved.

For the 7S/C Plasma Observatory, the seven-point magnetic measurements in 3-D yield $7 \times 3 = 21$ independent parameters, while the reference frame transformation provides nine constraints, resulting in $21+9=30$ input parameters in total. The objective is to determine the magnetic field (three parameters), first-order gradient (nine parameters), and quadratic magnetic gradient (18 parameters) at the mesocentre of the constellation, a total of $3+9+18=30$ parameters. Therefore, this scheme is reasonable such that the solution to the system of equations can be uniquely determined.

Clearly, the 9S/C magnetic measurements of HelioSwarm are sufficient to draw first-order and quadratic magnetic gradients using this method. These results indicate that the developed method is suitable for constellations comprising at least seven spacecraft.

### 2.2 Practical steps of the algorithm


Details of the steps used are given below.

### 2.2.1 The zeroth iteration:

We first assume a linear approximation in space and let $G_{ij}^{(0)} = 0$. The magnetic field $\mathbf{B}_c^{(0)}$ and its linear gradient

$(\nabla \mathbf{B})^{(0)}$ at the mesocentre of the constellation can then be obtained using the following formulas (Harvey, 1998; Shen et al., 2003):

$$B_{ci}^{(0)} = \frac{1}{N} \sum_{\alpha=1}^{N} B_{\alpha i} \,, \tag{4}$$

$$\left( \partial_i B_j \right)_c^{(0)} = \frac{1}{N} \sum_{\alpha=1}^{N} B_{\alpha i} r_{\alpha k} R_{kj}^{-1} \,. \tag{5}$$

where the volume tensor is $R_{kj} = \frac{1}{N} \sum_{\alpha=1}^{N} r_{\alpha k} r_{\alpha j}$ or $\mathbf{R} \equiv \frac{1}{N} \sum_{\alpha=1}^{N} \mathbf{r}_\alpha \mathbf{r}_\alpha$, where N is the number of spacecraft within the

constellation, and $R_{kj}^{-1}$ is the inverse of the volume tensor $R_{kj}$. The determinant of the volume tensor is required to be nonzero,

i.e., $R = \det \left( R_{kj} \right) \neq 0$. This is equivalent to that the constellation is non-planar (i.e., not all spacecraft are in the same plane).

The temporal variation rate $\left( \partial_t \mathbf{B} \right)_c^{(0)}$ is readily obtained from central differences of the magnetic observation time series.

Now the frame transformation relationships (3) are reduced to the apparent velocity $\mathbf{V}^{(0)}$ of the magnetic structure and the

longitudinal components of the quadratic magnetic gradient $\left( \partial_3 \nabla \mathbf{B} \right)^{(1)}$.

First, the zeroth approximation of the apparent velocity of the magnetic structure $\mathbf{V}^{(0)}$ can be found using the frame

transformation relationship:

$$\left( \partial_t \mathbf{B} \right)^{(0)} = -\mathbf{V}^{(0)} \cdot \left( \nabla \mathbf{B} \right)^{(0)} \,, \tag{6}$$

Then, using the relationship:

$$\partial_t \left( \nabla \mathbf{B} \right)^{(0)} = -\mathbf{V}^{(0)} \cdot \left( \nabla \nabla \mathbf{B} \right)^{(1)} \,, \tag{7}$$

the longitude components of the quadratic magnetic gradient at first order can be drawn as:

$$\left( \partial_3 \nabla \mathbf{B} \right)_c^{(1)} = \frac{1}{V^{(0)}} \partial_t \left( \nabla \mathbf{B} \left( t, \mathbf{r}_c \right) \right)^{(0)} \,, \tag{8}$$

which are just $\left( G_{31}^{(1)}, G_{32}^{(1)}, G_{33}^{(1)} \right)$.

The remaining components of the quadratic magnetic gradients can be deduced using the least-squares method.

Assuming that:

$$S = \frac{1}{N}\sum_{\alpha=1}^{N}\left[ f_c^{(0)} + x_{(\alpha)}^i g_i^{(0)} + \frac{1}{2}x_{(\alpha)}^i x_{(\alpha)}^j G_{ij}^{(1)} - f_{(\alpha)} \right]^2, \tag{9}$$

which can also be written as:

$$S = \frac{1}{N}\sum_{\alpha=1}^{N}\left[ f_c^{(0)} + x_{(\alpha)}^i g_i^{(0)} - f_{(\alpha)} + \left(1-\frac{1}{2}\delta_{i3}\right)x_{(\alpha)}^i x_{(\alpha)}^3 G_{i3}^{(1)} + \frac{1}{2}x_{(\alpha)}^p x_{(\alpha)}^q G_{pq}^{(1)} \right]^2. \tag{10}$$

where, $p,q = 1,2$.

If $\delta S = 0$, then

$$\frac{\partial S}{\partial G_{pq}} = \frac{1}{N}\sum_{\alpha=1}^{N}2\left[ f_c^{(0)} + x_{(\alpha)}^i g_i^{(0)} - f_{(\alpha)} + \left(1-\frac{1}{2}\delta_{i3}\right)x_{(\alpha)}^i x_{(\alpha)}^3 G_{i3}^{(1)} + \frac{1}{2}x_{(\alpha)}^r x_{(\alpha)}^s G_{rs}^{(1)} \right]\cdot x_{(\alpha)}^p x_{(\alpha)}^q = 0. \tag{11}$$

which reduces to:

$$f_c^{(0)}\sum_{\alpha=1}^{N}x_{(\alpha)}^p x_{(\alpha)}^q + \sum_{\alpha=1}^{N}x_{(\alpha)}^i x_{(\alpha)}^p x_{(\alpha)}^q g_i^{(0)} - \sum_{\alpha=1}^{N}f_{(\alpha)}x_{(\alpha)}^p x_{(\alpha)}^q + \sum_{\alpha=1}^{N}\left(1-\frac{1}{2}\delta_{i3}\right)x_{(\alpha)}^i x_{(\alpha)}^3 x_{(\alpha)}^p x_{(\alpha)}^q G_{i3}^{(1)}$$
$$+ \frac{1}{2}\sum_{\alpha=1}^{N}x_{(\alpha)}^r x_{(\alpha)}^s x_{(\alpha)}^p x_{(\alpha)}^q G_{rs}^{(1)} = 0 \tag{12}$$

Resulting in $G_{rs}^{(1)}\left(r,s=1,2\right)$, i.e., $\left(G_{21}^{(1)},G_{22}^{(1)},G_{11}^{(1)}\right)$. The constellation must be nonplanar to achieve this result. This is verified as follows.

Following Zhou & Shen (2024), in order for the solution to exist, it is expected that the position of all the spacecraft in the constellation must not obey the following formula

$$a_{11}\left(x_{(\alpha)}^1\right)^2 + a_{12}x_{(\alpha)}^1 x_{(\alpha)}^2 + a_{12}x_{(\alpha)}^2 x_{(\alpha)}^1 + a_{22}\left(x_{(\alpha)}^2\right)^2 = 0, \tag{13}$$

where $a_{rs}\left(r,s=1,2\right)$ are fixed coefficients. The above equations can be rewritten as

$$a_{11}\left(x_{(\alpha)}^1 / x_{(\alpha)}^2\right)^2 + 2a_{12}\left(x_{(\alpha)}^1 / x_{(\alpha)}^2\right) + a_{22} = 0, \tag{14}$$

which reduce to $x^1_{(\alpha)} / x^2_{(\alpha)} = constant$. It means that all the spacecraft are in the plane parallel to the $x_3$ axis or the motion direction. Therefore, it is necessary that the constellation should not be planar in order to deduce the quadratic magnetic gradients as well as the linear magnetic gradient. The next iterations would also require this condition.

**2.2.2 First order iteration**

Assuming that:

$$S = \frac{1}{N} \sum_{\alpha=1}^{N} \left[ f_c^{(1)} + x^i_{(\alpha)} g_i^{(1)} + \frac{1}{2} x^i_{(\alpha)} x^j_{(\alpha)} G_{ij}^{(1)} - f_{(\alpha)} \right]^2 . \tag{15}$$

If $\delta S = 0$, then:


$$\frac{\partial S}{\partial f_c^{(1)}} = 0 , \quad \frac{\partial S}{\partial g_i^{(1)}} = 0 . \tag{16}$$

From $\dfrac{\partial S}{\partial f_c^{(1)}} = 0$, it can be assumed that:

$$\frac{1}{N} \sum_{\alpha=1}^{N} \left[ f_c^{(1)} + x^i_{(\alpha)} g_i^{(1)} + \frac{1}{2} x^i_{(\alpha)} x^j_{(\alpha)} G_{ij}^{(1)} - f_{(\alpha)} \right] = 0 . \tag{17}$$


Meaning that:

$$f_c^{(1)} = \frac{1}{N} \sum_{\alpha=1}^{N} f_{(\alpha)} - \frac{1}{2N} \sum_{\alpha=1}^{N} x^i_{(\alpha)} x^j_{(\alpha)} G_{ij}^{(1)} = \frac{1}{N} \sum_{\alpha=1}^{N} f_{(\alpha)} - \frac{1}{2} R^{ij} G_{ij}^{(1)} . \tag{18}$$

If $\dfrac{\partial S}{\partial g_i^{(1)}} = 0$, this can be reduced to:


$$\frac{1}{N} \sum_{\alpha=1}^{N} \left[ f_c^{(1)} + x^i_{(\alpha)} g_i^{(1)} + \frac{1}{2} x^i_{(\alpha)} x^j_{(\alpha)} G_{ij}^{(0)} - f_{(\alpha)} \right] x^k_{(\alpha)} = 0 , \tag{19}$$

i.e.,

$$\frac{1}{N}\sum_{\alpha=1}^{N} x_{(\alpha)}^{k} x_{(\alpha)}^{i} g_{i}^{(1)} + \frac{1}{2}\frac{1}{N}\sum_{\alpha=1}^{N} x_{(\alpha)}^{k} x_{(\alpha)}^{i} x_{(\alpha)}^{j} G_{ij}^{(0)} - \frac{1}{N}\sum_{\alpha=1}^{N} f_{(\alpha)} x_{(\alpha)}^{k} = 0. \tag{20}$$

The tensor $R^{kij} = \dfrac{1}{N}\sum_{\alpha=1}^{N} x_{(\alpha)}^{k} x_{(\alpha)}^{i} x_{(\alpha)}^{j}$ is then defined, resulting in:

$$R^{ki} g_{i}^{(1)} + \frac{1}{2} R^{kij} G_{ij}^{(0)} - \frac{1}{N}\sum_{\alpha=1}^{N} f_{(\alpha)} x_{(\alpha)}^{k} = 0. \tag{21}$$

Therefore, the first magnetic gradient is:

$$g_{\ell}^{(1)} = -\frac{1}{2}\left(R^{-1}\right)^{k\ell} R^{kij} G_{ij}^{(0)} + \left(R^{-1}\right)^{k\ell} \cdot \frac{1}{N}\sum_{\alpha=1}^{N} f_{(\alpha)} x_{(\alpha)}^{k}. \tag{22}$$

Using equation (3), it is now possible to obtain the corrected apparent velocity $\mathbf{V}^{(1)}$ of the magnetic structure and the longitudinal components of the corrected quadratic magnetic gradient $\left(\partial_{3}\nabla\mathbf{B}\right)^{(2)} \left(\left(\partial_{3}\partial_{i}\mathbf{B}\right)^{(2)}\right)$, as in the zeroth iteration.

The least-squares method is then used to obtain the remaining nine components of the corrected quadratic magnetic gradient. If:

$$\begin{aligned}
S &= \frac{1}{N}\sum_{\alpha=1}^{N}\left[ f_{(c)}^{(1)} + x_{(\alpha)}^{i} g_{i}^{(1)} + \frac{1}{2} x_{(\alpha)}^{i} x_{(\alpha)}^{j} G_{ij}^{(2)} - f_{(\alpha)} \right]^{2} \\
&= \frac{1}{N}\sum_{\alpha=1}^{N}\left[ f_{c}^{(1)} + x_{(\alpha)}^{i} g_{i}^{(1)} - f_{(\alpha)} + \left(1 - \frac{1}{2}\delta_{i3}\right) x_{(\alpha)}^{i} x_{(\alpha)}^{3} G_{i3}^{(2)} + \frac{1}{2} x_{(\alpha)}^{p} x_{(\alpha)}^{q} G_{pq}^{(2)} \right]^{2}
\end{aligned} \tag{23}$$

then $G_{pq}^{(2)} \left(p, q = 1, 2\right)$ can be obtained using the same procedure as that used for the zeroth iteration. So that all the components of the corrected quadratic magnetic gradient $\left(\nabla\nabla\mathbf{B}\right)^{(2)}$ are obtained.

Similarly, two or more iterations can be performed until stable linear and second-order magnetic gradients are obtained.

This algorithm requires that the constellation be composed of at least seven spacecraft and that its configuration is non-planar. Because both the 9S/C HelioSwarm and 7S/C Plasma Observatory satisfy these requirements, the linear and quadratic magnetic gradients can be readily obtained.

The Curlometer technique (Dunlop et al., 2002b) is used to calculate the current density based on multiple spacecraft magnetic measurements, with the relative error estimated by the ratio between the divergence and curl of the magnetic field, i.e., $\left|\dfrac{\nabla\cdot\mathbf{B}}{\nabla\times\mathbf{B}}\right|$. If the length and the magnetic field are normalized by the characteristic distance and magnetic strength

$\left(D, B_0\right)$ , the equation becomes $\left|\dfrac{\overline{\overline{\nabla}} \cdot \overline{\mathbf{B}}}{\overline{\overline{\nabla}} \times \overline{\mathbf{B}}}\right| \approx \left|\dfrac{\overline{\nabla} \cdot \overline{\mathbf{B}}}{1}\right| = \left|\overline{\nabla} \cdot \overline{\mathbf{B}}\right|$ . Therefore, the dimensionless divergence of the magnetic field

calculated with observation data can be regarded a reasonable measure of the relative error within the linear magnetic gradient.

Similarly, the dimensionless $\left|\overline{\nabla}\left(\overline{\nabla} \cdot \overline{\mathbf{B}}\right)_c\right|$ can be used as a measure describing the relative error in the quadratic magnetic

gradient derived using the method.

## 3. Comparison of new method with analytical modelling

In this section, the new method is applied to two analytical magnetic field models (a cylindrical force-free flux rope

and a dipole magnetic field) to evaluate its validity and accuracy. The applicability of this approach was tested on the 7S/C

Plasma Observatory (N=7) under the assumption that the seven-spacecraft cluster crosses a magnetic field structure (as

illustrated in Figure 1) by comparing the linear and quadratic gradients of the magnetic field obtained by the new method with

those obtained by accurate modelling.

The positions of the seven spacecrafts in the barycentric coordinate system were generated randomly with Cartesian

coordinates between -0.02 and 0.02 $R_E$, as seen in Table 1. The 7S/C array is illustrated in Figure 2.

**Table 1. Coordinates of the seven spacecraft in the barycentre coordinate system, with $\alpha$ denoting spacecraft number.**

| $\alpha$ | $x_{(\alpha)}$ $(R_E)$ | $y_{(\alpha)}$ $(R_E)$ | $z_{(\alpha)}$ $(R_E)$ |
|---|---|---|---|
| 1 | 0.0105 | 0.0016 | 0.0100 |
| 2 | 0.0135 | 0.0153 | -0.0119 |
| 3 | -0.0124 | 0.0155 | -0.0026 |
| 4 | 0.0138 | -0.0114 | 0.0139 |
| 5 | 0.0044 | 0.0157 | 0.0097 |
| 6 | -0.0134 | 0.0152 | 0.0153 |
| 7 | -0.0074 | -0.0005 | 0.0052 |

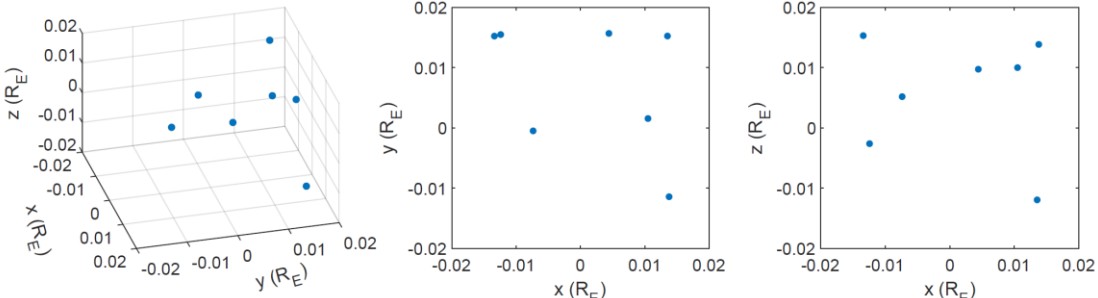

**Figure 2. Configuration of the 7-S/C constellation.**

The characteristic configuration of the spacecraft is described using several parameters. The three eigenvalues of the volumetric tensor $R^{ij}$ are represented by $w_1$, $w_2$, and $w_3$ (where $w_1 \geq w_2 \geq w_3$) (Harvey, 1998), with their square roots representing the characteristic half-widths of the S/C in the three orthogonal directions along the corresponding eigenvectors (Harvey, 1998). The characteristic size of the S/C constellation is twice the square root of the maximum eigenvalue,

$L = 2\sqrt{w_1}$ (Robert et al., 1998; Shen et al, 2012). For the 7-S/C constellation tested in this section, the three eigenvalues are

$$w_1 = 0.1643 \times 10^{-3} R_E^2 \; , \quad w_2 = 0.1104 \times 10^{-3} R_E^2 \; , \quad \text{and} \quad w_3 = 0.0341 \times 10^{-3} R_E^2 \; . \quad \text{The characteristic size is}$$

$$L = 2\sqrt{w_1} = 0.0256 \, R_E = 163.33 \, \text{km} .$$

**3.1 Flux Rope**

The flux rope was assumed to be force-free and cylindrically symmetrical. The magnetic field of the flux rope can be described using the Helmholtz equation, for which Lundquist (1950) provided analytical solutions in terms of the Bessel functions.

$$B_r = 0, \; B_\varphi(r) = B_0 J_1(\alpha r), \; B_z(r) = B_0 J_0(\alpha r), \tag{24}$$

where $r$ is the radial distance from the centric axis; $\alpha$ is a constant, with $1/\alpha$ representing the characteristic scale of the flux rope; $B_0$ is the peak axial field intensity; and $J_0$ and $J_1$ are the zeroth- and first-order Bessel functions of the first kind, respectively. For this test, we set $B_0$=60 nT and $\alpha$=1/$R_E$.

The 7-S/C array was assumed to cross the flux rope in a straight line at uniform velocity. The array is represented by the barycentre with the red dot in Figure 3, and moves from (-2, 0, 0) $R_E$ to (2, 0, 0) $R_E$ along the x-axis over a time interval of

100 s. The resolution of the magnetic field measurement was set to 1 s for the time-series observations, and the characteristic

size of the 7-S/C array was set to $L=0.0256\ R_E$ for the gradients of the magnetic field at the barycentre along the trajectory to be obtained.

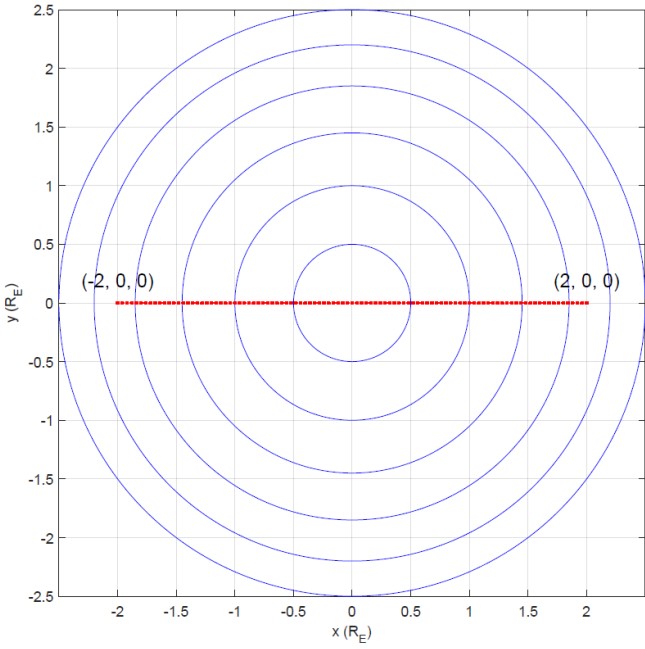

**Figure 3. The cylindrical force-free flux rope crossing by the 7-S/C constellation as viewed from the axial direction. Trajectory of the barycentre of the constellation from (-2, 0, 0) $R_E$ to (2, 0, 0) $R_E$ over 100 s is shown by the red dotted line. Blue lines represent magnetic field lines.**

The linear gradient of the magnetic field ($\nabla_i B_k$) has nine components, while the quadratic gradients ($\nabla_i \nabla_j B_k$) comprise 27 components. According to the analytical flux-rope model and symmetry of the quadratic gradients, only five independent components of the quadratic gradients $\nabla_1 \nabla_2 B_1$, $\nabla_1 \nabla_1 B_2$, $\nabla_2 \nabla_2 B_2$, $\nabla_1 \nabla_1 B_3$, and $\nabla_2 \nabla_2 B_3$, and three components of the linear gradients $\nabla_2 B_1$, $\nabla_1 B_2$, and $\nabla_1 B_3$ are nonzero points on the *x*-axis when using Cartesian coordinates, simplifying the comparison between the gradients derived from the proposed method and the analytical model.

The impact of iteration on the results was investigated first, with the results at two different points used to demonstrate the variation in the relative errors under iteration, as illustrated in Figure 4. The relative error is defined as $\left| \left( X_{algorithm} - X_{accurate} \right) \middle/ X_{accurate} \right| \times 100\%$, where $X_{algorithm}$ and $X_{accurate}$ represent the algorithm gradients derived using the new method and accurate values from the analytical model at the barycentre, respectively. As shown in Figure 4(a) and 4(c), the linear gradients converged to certain values within 50 iterations, and the final relative errors were less than 0.02%. Figure 4(b) and 4(d) also indicate that the quadratic gradients converge. However, some quadratic gradients converged faster than others with fewer relative errors, and final relative errors of no more than 1.5% were obtained after 100 iterations. The maximum number of iterations was set to 100; thus, the gradients could be derived with good accuracy.

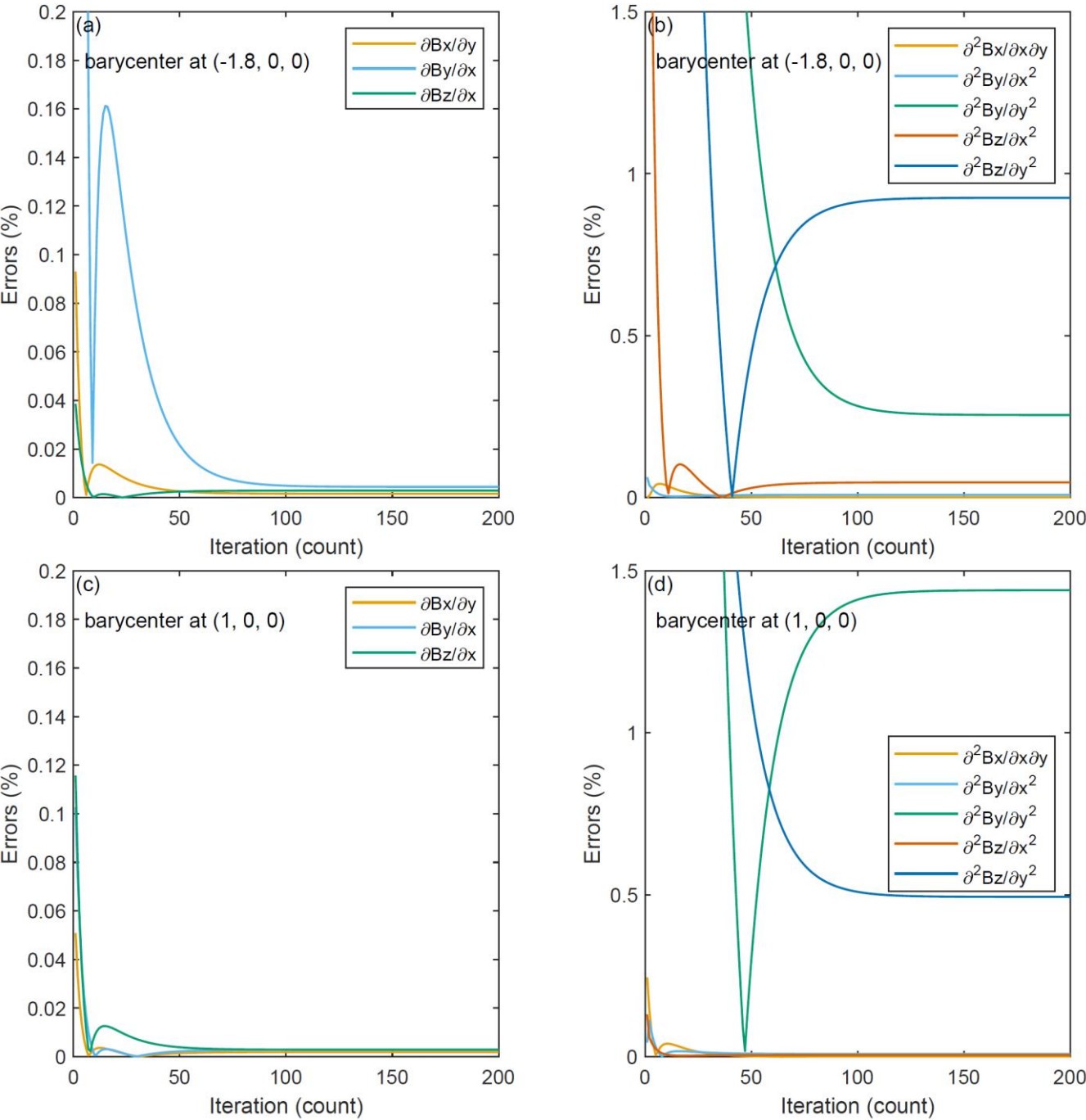

**Figure 4. Relative errors in the nonzero components of the linear and quadratic gradients with different iteration numbers at various barycentres.**

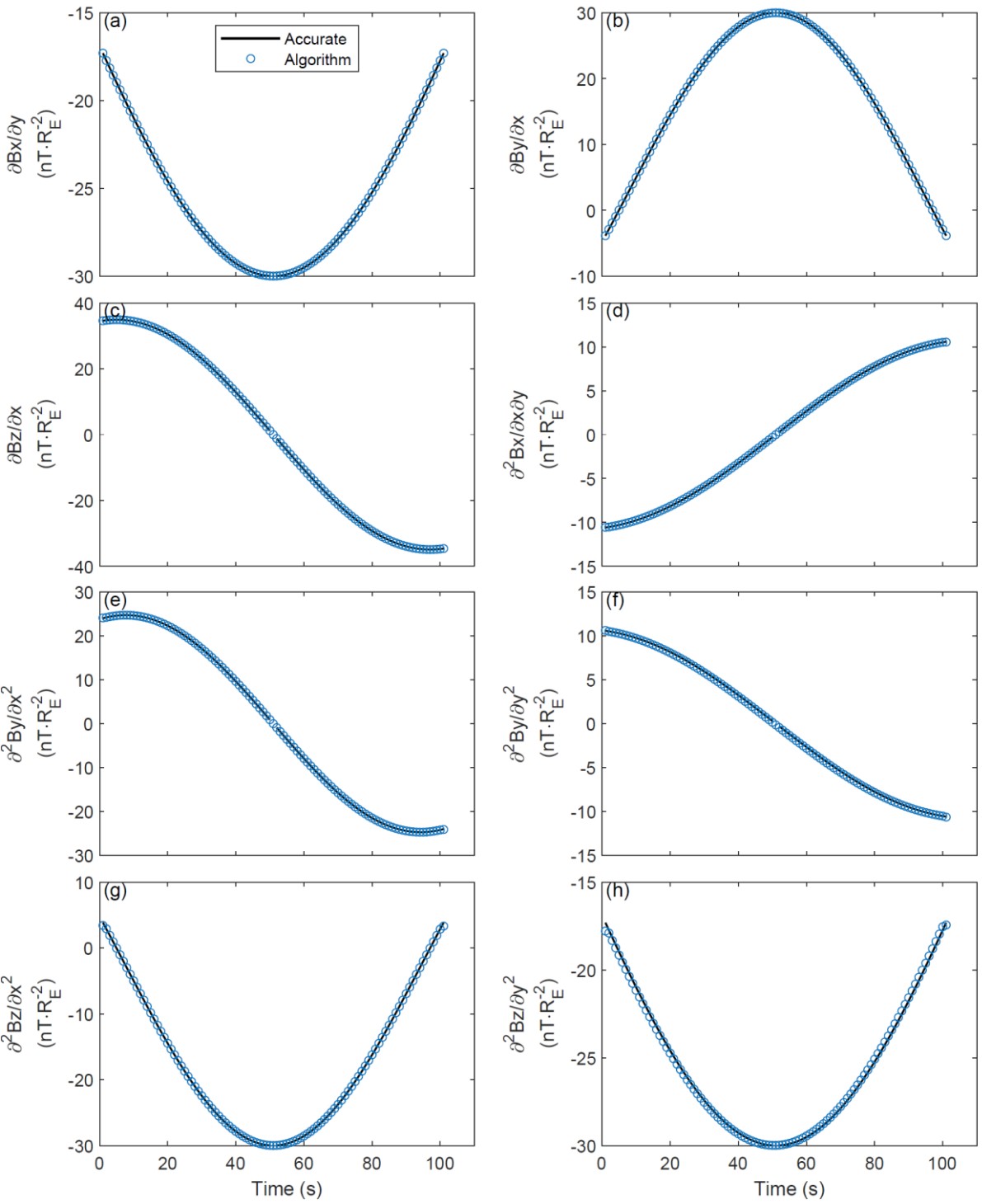

**Figure 5. Time series showing nonzero components of the linear and quadratic gradients. Circles and solid lines represent the results obtained using the algorithm and accurate modelling, respectively.**


Figure 5 shows a comparison of the nonzero linear and quadratic gradients at the barycentre derived from our method with those derived from the analytical model. The algorithm gradients are consistent with the accurate gradients, indicating that the proposed method is effective and precise when used with flux ropes.

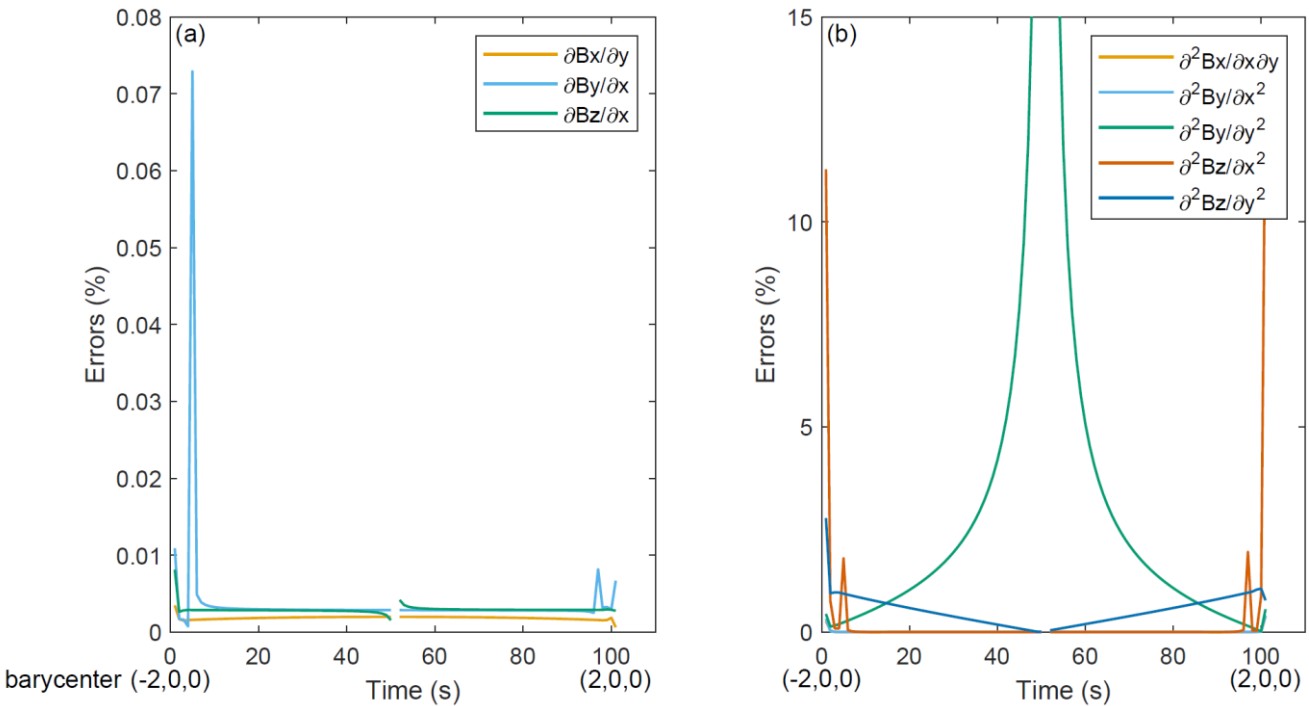


**Figure 6. Relative errors in the nonzero components of the linear and quadratic gradients along the crossing path.**

The relative errors of the gradients at points along the trajectory are shown in Figure 6. All the relative errors of the linear gradients were less than 0.1%, and the vast majority of the relative errors for the quadratic gradients did not exceed 5%.

It should be noted that the barycentre is at (0,0,0) at 50 s and that the nonzero components of the linear and quadratic gradients do not exist at (0,0,0). The barycentre is at (-0.04,0,0) $R_E$ at 49 s, when accurate modelling and algorithm values for the quadratic gradient $\nabla_2\nabla_2 B_2$ are 0.3 and 0.1570 $\mathrm{nT}\cdot\mathrm{R}_E^2$, respectively. The relative error approaches 50%; however, the absolute error is just 0.143 $\mathrm{nT}\cdot\mathrm{R}_E^2$, which is approaching zero. Symmetrically, the situation described is the same as it would be if the barycentre were at (0.04,0,0).


## 3.2 Dipole field

The proposed method was also tested and verified using a magnetic dipole field. The geomagnetic dipole field is mathematically expressed as:

$$\begin{cases} B_x = -\dfrac{3xz}{r^5} B_0 \\[2mm] B_y = -\dfrac{3yz}{r^5} B_0 \\[2mm] B_z = \dfrac{r^2 - 3z^2}{r^5} B_0 \end{cases}, \tag{25}$$

where $B_0$ is the magnetic field at the Earth's equator and is defined by $B_0 = \dfrac{\mu_0 M}{4\pi R_E^3} = 30008 \text{ nT}$ ; $M = 7.76 \times 10^{22} \text{ A} \cdot \text{m}^2$

is the geomagnetic moment, with its direction set anti-parallel to the z-axis; $x$, $y$ and $z$ are the coordinates of the field points measured by $R_E$, and $r = \sqrt{x^2 + y^2 + z^2}$ is the radial distance from the origin measured by $R_E$.

The 7-S/C array was assumed to cross the dipole field in a straight line at constant velocity, with the barycentre parallel to the $x$-axis and moving from (-5, 0, 5) $R_E$ to (5, 0, 5) $R_E$ over 125 s, as illustrated in Figure 7. The resolution of the magnetic field measurement was set to 1 s and the characteristic size of the 7-S/C array was set to 0.0256 $R_E$, which is the same as that of the flux-rope case, for the gradients of the magnetic field at the barycentre along the trajectory to be obtained.

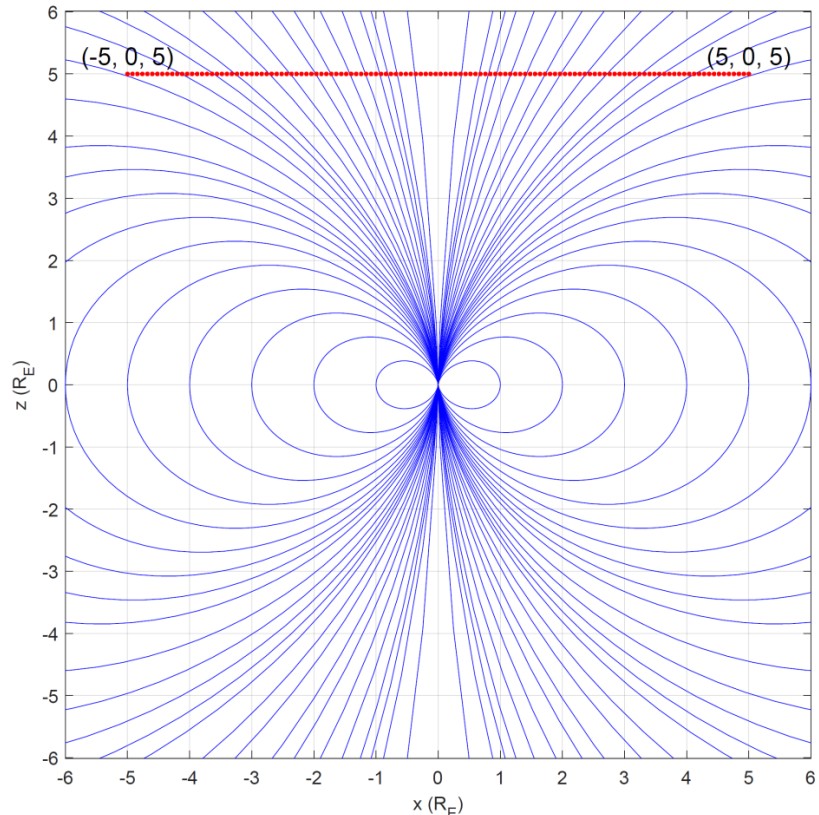

**Figure 7. The magnetic dipole field crossed by the 7-S/C array. Trajectory of the barycentre of the 7-S/C array is from (-5, 0, 5) $R_E$ to (5, 0, 5) $R_E$ over 126 s as shown by the red dotted line. Blue lines represent magnetic field lines.**

Only nonzero independent components are displayed, similar to the flux rope case. In view of the mathematical expression of the dipole field, ten independent components of the quadratic gradients and four independent components of the linear gradients were nonzero along the crossing path.

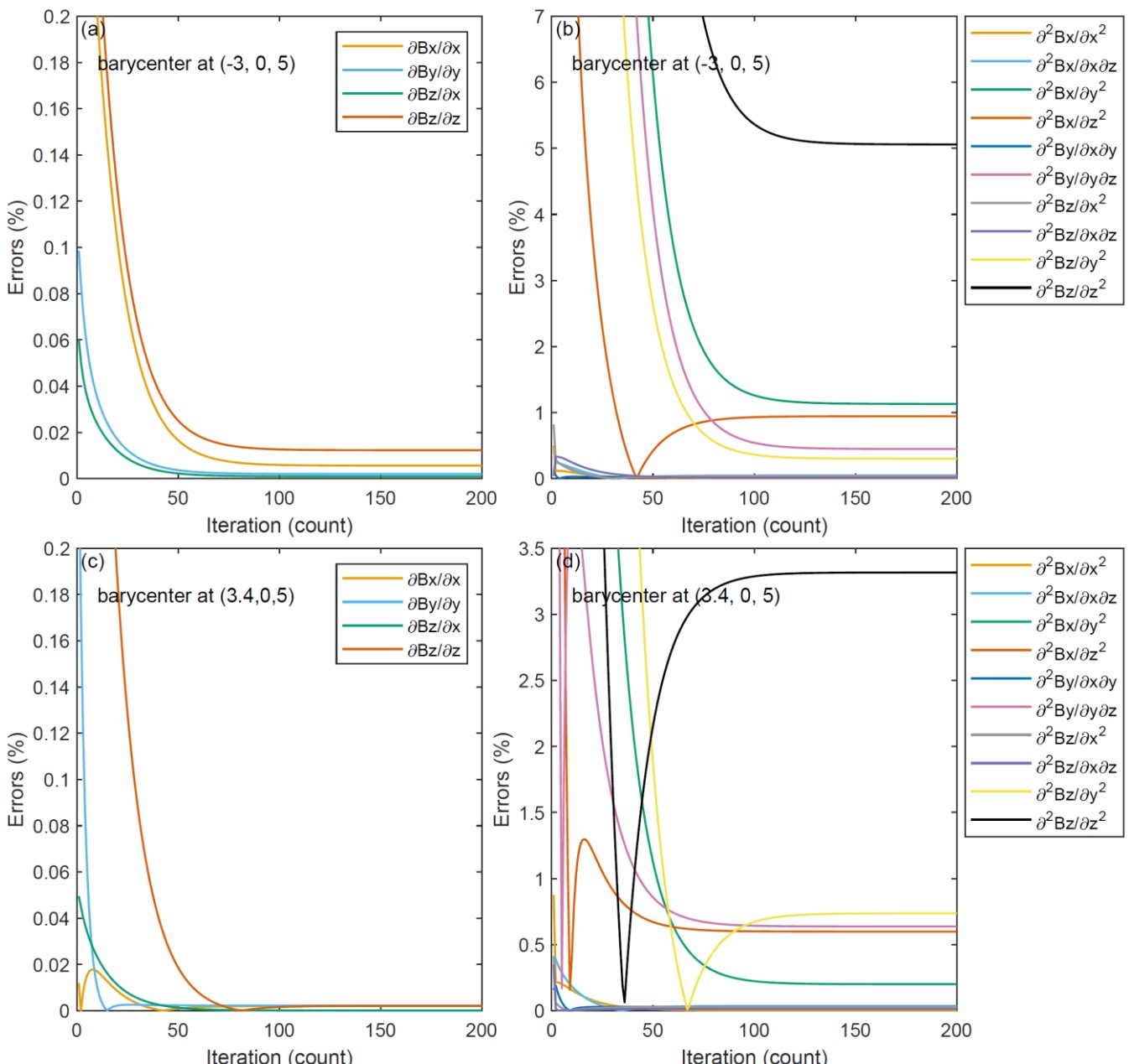

**Figure 8. Relative errors in the nonzero components of the linear and quadratic gradients with different iteration numbers at various barycentres.**

Figure 8 shows the variation in the relative errors under iteration at two different points. As shown in Figure 8(a) and 8(c), the linear gradients converged to certain values within 60 iterations, with final relative errors of less than 0.02%. Figure 8(b) and 8(d) show that the quadratic gradients also converge to low errors. After 100 iterations, most of the relative errors of

the quadratic gradients were less than 1%, and the largest relative error was no more than 6%. These results suggest that it is reasonable to set the maximum number of iterations to 100 for the gradients to be derived with good accuracy in this case.

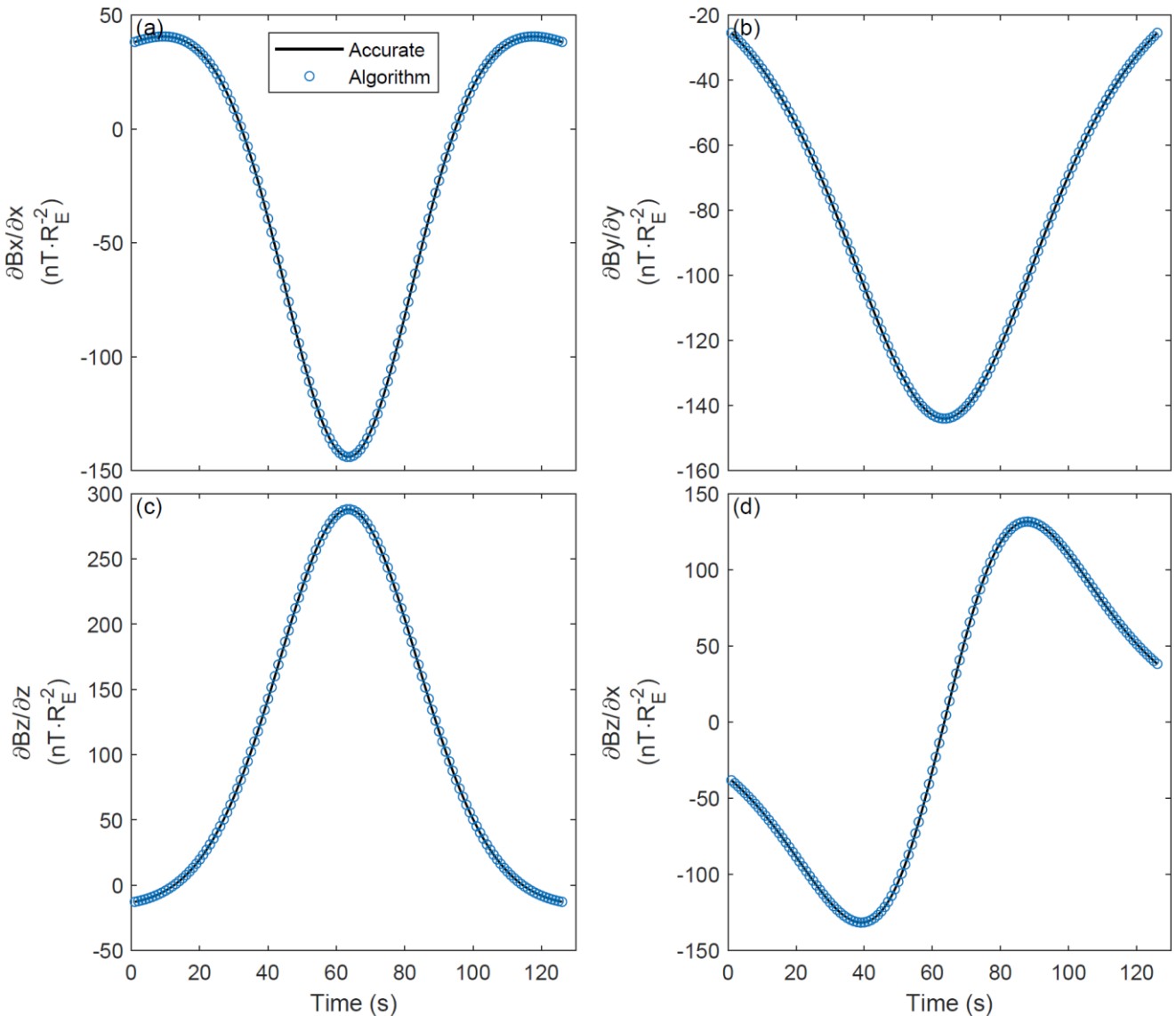

**Figure 9. Time series showing nonzero components of the linear gradients. Circles and solid lines represent the results obtained using the algorithm and accurate modelling, respectively.**

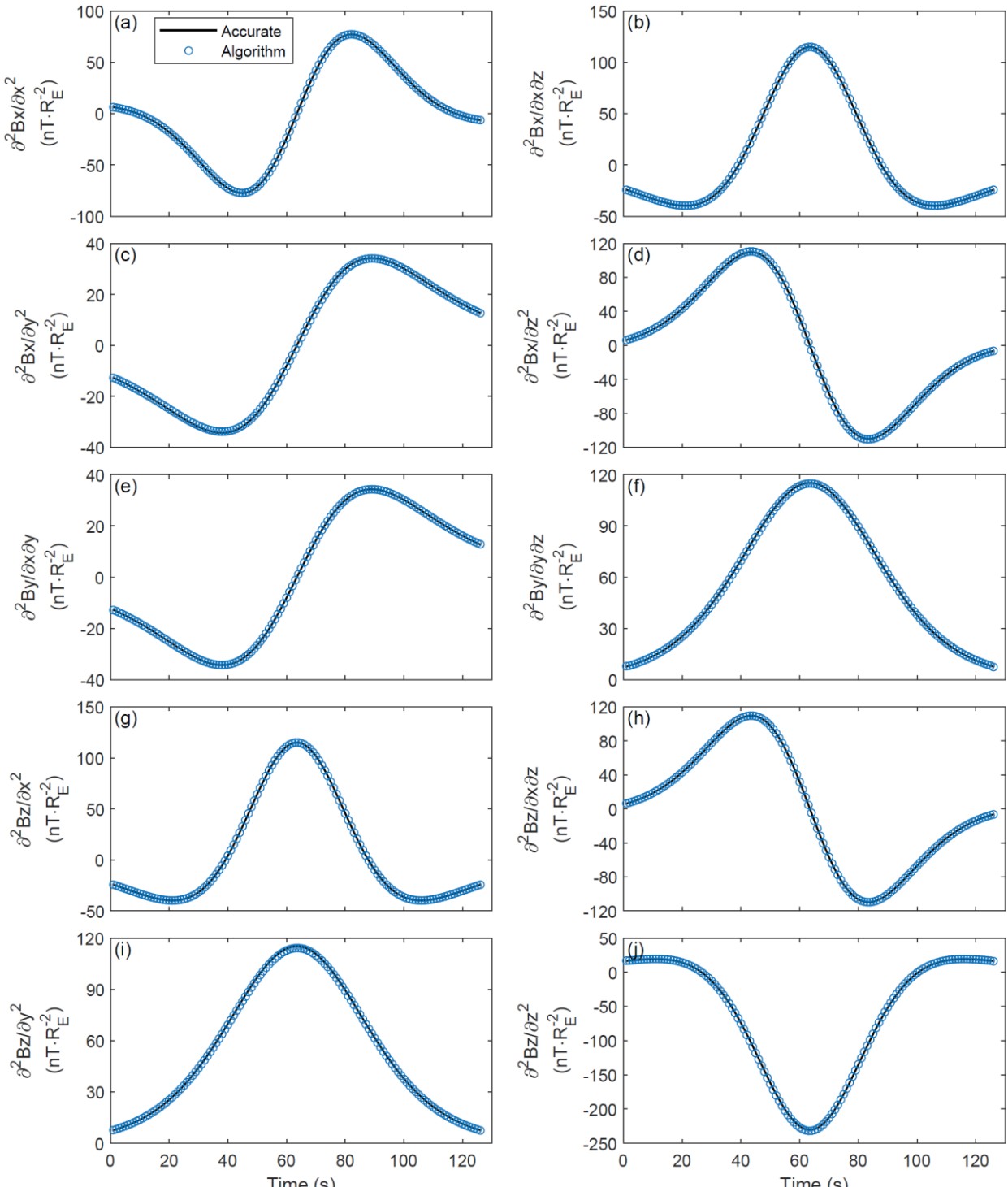

**Figure 10. Time series showing nonzero components of the quadratic gradients. Circles and solid lines represent the results obtained using the algorithm and accurate modelling, respectively.**

Figure 9 shows a comparison of the nonzero linear gradients derived from our method with those derived from the analytical model. A comparison of the nonzero quadratic gradients derived from the different sources is shown in Figure 10. Both Figure 9 and 10 indicate that the algorithm gradients are entirely consistent with those obtained from the accurate model, suggesting that the developed method is effective and precise for use with the dipole field.


Figure 11. Relative errors in the nonzero components of the linear and quadratic gradients along the crossing path.

Figure 11 shows the relative errors of the gradients at the measured points along the crossing path. All the relative errors for the linear gradients were less than 0.25 %, and most of the relative errors in the quadratic gradients were less than 5%. It should be noted that the barycentre is at (2.92, 0, 5) $R_E$ at 100s, and the accurate and algorithm quadratic gradients $\nabla_3 \nabla_3 B_3$ are -1.2584 and -0.6461 $nT \cdot R_E^2$, respectively. The relative error approaches 50%; however, the absolute error is 0.6123 $nT \cdot R_E^2$, which is approaching zero. The barycentre is at (-0.04, 0, 5) $R_E$ at 63 s and (0.04, 0, 5) $R_E$ at 64 s. Similarly, the absolute errors in the quadratic gradients $\nabla_2 \nabla_2 B_1$ and $\nabla_3 \nabla_3 B_1$ were no more than 1 $nT \cdot R_E^2$, whereas the relative errors were approximately 30%.

### 3.3 Discussion

The two analytical magnetic field models (cylindrical force-free flux rope and dipole magnetic field) are simplified and highly symmetrical structures. The linear gradient of the magnetic field has 9 components, while the quadratic gradients comprise 18 independent components due to the symmetry of quadratic gradients. For the flux rope case, only 3 components of linear gradient and 5 components of quadratic gradients have been assessed. But for dipole field case, 4 components of linear gradient and 10 components of quadratic gradients have been assessed. The number of assessed parameters has reached half. However, only a subset of the 9+18=27 components can be assessed. In this study, we have chosen a symmetric model magnetic field in order to easily compare our results with the analytic calculations. The zero components of the magnetic gradients are calculated with the algorithm and checked. Further evaluation of the algorithm with a less symmetric magnetosphere model could be useful.

### 4. Errors

In this Section, we consider the diverse sources of errors, namely the truncation error, discretisation error, iteration error, and measurement error or random error, that can impact the linear and quadratic magnetic spatial gradient estimation. We introduce and discuss them as follows. Further detailed analyses can be found in Appendix A-C.

Discretisation errors arise from the spatial resolution of measurement, which is the combined effect of finite temporal resolution and the relative motion of a probe with respect to the magnetic structure during measurement period. The significance of these errors can be accessed by comparing the spatial resolution---defined as the distance a probe travels during a single sampling---with the separation between probes, over which the measured data are subtracted from one another. Typically, spacecraft separation within a constellation ranges from several 100km to several 1000km, while the temporal resolution of magnetic measurement $\Delta t$ is about 0.01 s, i.e. $\Delta t = 0.01 \text{s}$. Assuming a magnetic structure moves at a velocity V<500 km/s relative to the spacecraft, the spatial resolution along the motion direction is about $v\Delta t < 5km$, which is significantly smaller than the S/C separation. Therefore, the corresponding discretisation errors are expected to be small.

The iterative method provides converging solution with decreasing errors as the number of iterations increases as demonstrated in Section 3. A particular type of error is the mismatch between the actual limit of the procedure and the approximation reached after a finite number of iterations. This error may be termed iteration error. We note that it is not associated with the finite resolution of the spacecraft array or the time series. As shown in Figure 4, the iterative procedures help reduce the calculation error and make the calculation more stable. In addition, once the number of iterations is sufficient (e.g., above 100), the iterative error becomes rather insignificant (see Appendix A). More advanced evaluation on the iteration accuracy of this algorithm can be made in the future when the real mission data are available.

Due to noise or measurement inaccuracies of input data, the estimated parameters (first and second derivatives) will be subject to random errors. The noise or disturbances in the data can come from the measurement error or the presence of

high-frequency (physical) fluctuations such as those from plasma waves, which can make the calculation of the high order magnetic gradients very difficult (Shen et al. 2021a). Indeed, the relative errors of the gradients associated to the measurement errors are expected to be the same magnitude as the measurement errors (see Appendix C). When analysing the actual observation data, filtering methods should be employed to remove the high frequency components and avoid the negative effect of the noise. This process would help to extract large-scale magnetic structures under the consideration.

As discussed above, the discretisation error, iteration error, and measurement error are expected to be rather small for the magnetic configuration considered. In the following, only truncation error has been evaluated.

In Section 3, relative error is used to evaluate the truncation error of the proposed method. However, in some cases, evaluation with the relative error is not effective, while the gradient obtained from the accurate model is very small. Furthermore, the truncation error was evaluated under divergence-free magnetic field conditions.

Theoretically, the divergence of the magnetic field and the gradient of the magnetic field divergence are both exactly zero, as shown by $\nabla \cdot \boldsymbol{B} = 0$ and $\left|\nabla(\nabla \cdot \boldsymbol{B})\right| = 0$. To offer a uniform standard for evaluation, the divergence and gradient of divergence were non-dimensionalized with the corresponding characteristic quantity. The length was calibrated with the spatial characteristic scale of the magnetic structure $D$, and the magnetic field was calibrated with the characteristic magnetic field at the barycentre $B_c$. Therefore, two evaluation indices were introduced, represented by $\left(\overline{\nabla} \cdot \overline{\boldsymbol{B}}\right)_c$ and $\left|\overline{\nabla}\left(\overline{\nabla} \cdot \overline{\boldsymbol{B}}\right)_c\right|$. The values of the two indices can be used to evaluate the accuracies of the linear and quadratic gradients derived using the proposed method. Nevertheless, this evaluation is not perfect because it cannot include all partial components of the magnetic gradients (the formula $\nabla \cdot \boldsymbol{B} = 0$ contains 3 of the total 9 components of $\nabla \boldsymbol{B}$ while $\left|\nabla(\nabla \cdot \boldsymbol{B})\right| = 0$ contains 9 of the total 18 components of $\nabla\nabla\boldsymbol{B}$). The advantage to use them as the measures of the errors of the magnetic gradients is that they are robust and simple. We still have not found other better ways for evaluating the accuracy of the algorithm because the actual values of the magnetic gradients are unknown when analysing the real observation data.

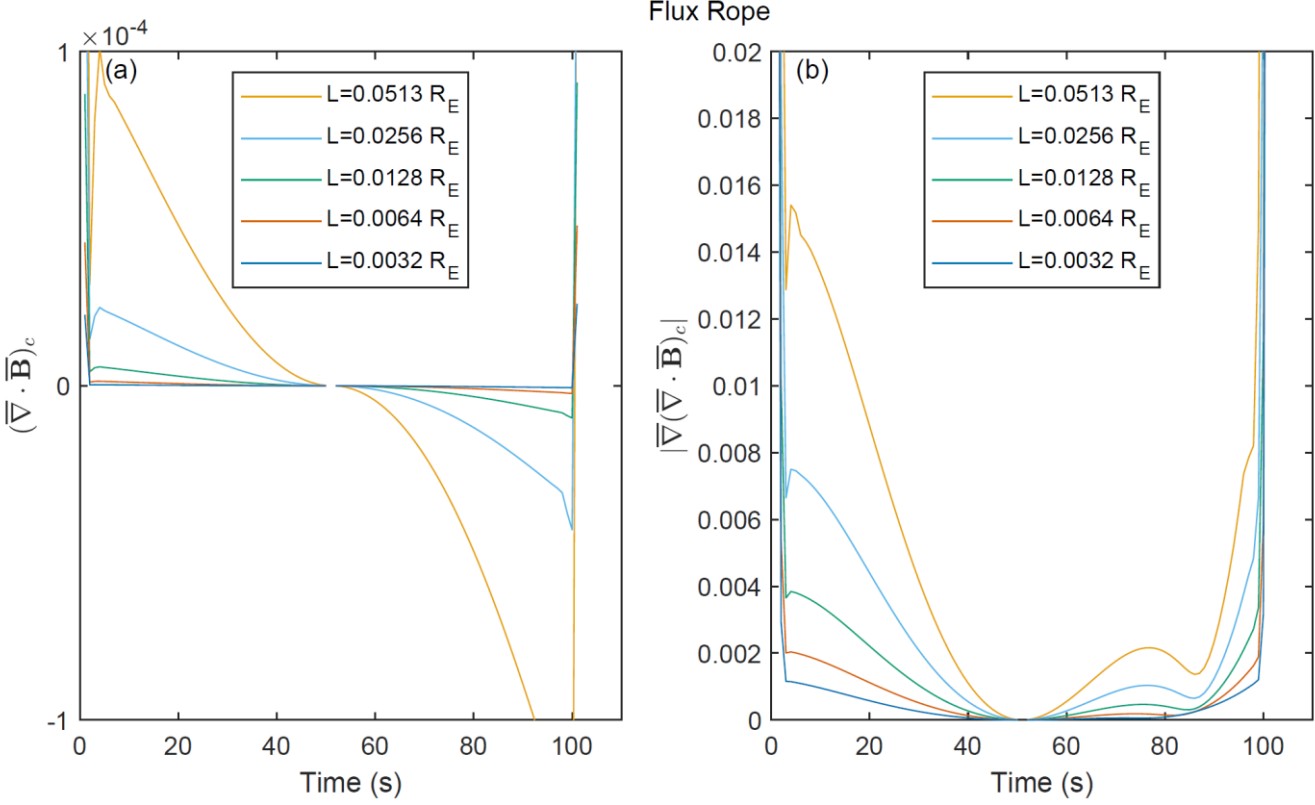

**Figure 12. Dimensionless divergence and gradient of divergence for magnetic field along the flux rope crossing path with different characteristic S/C sizes (L).**

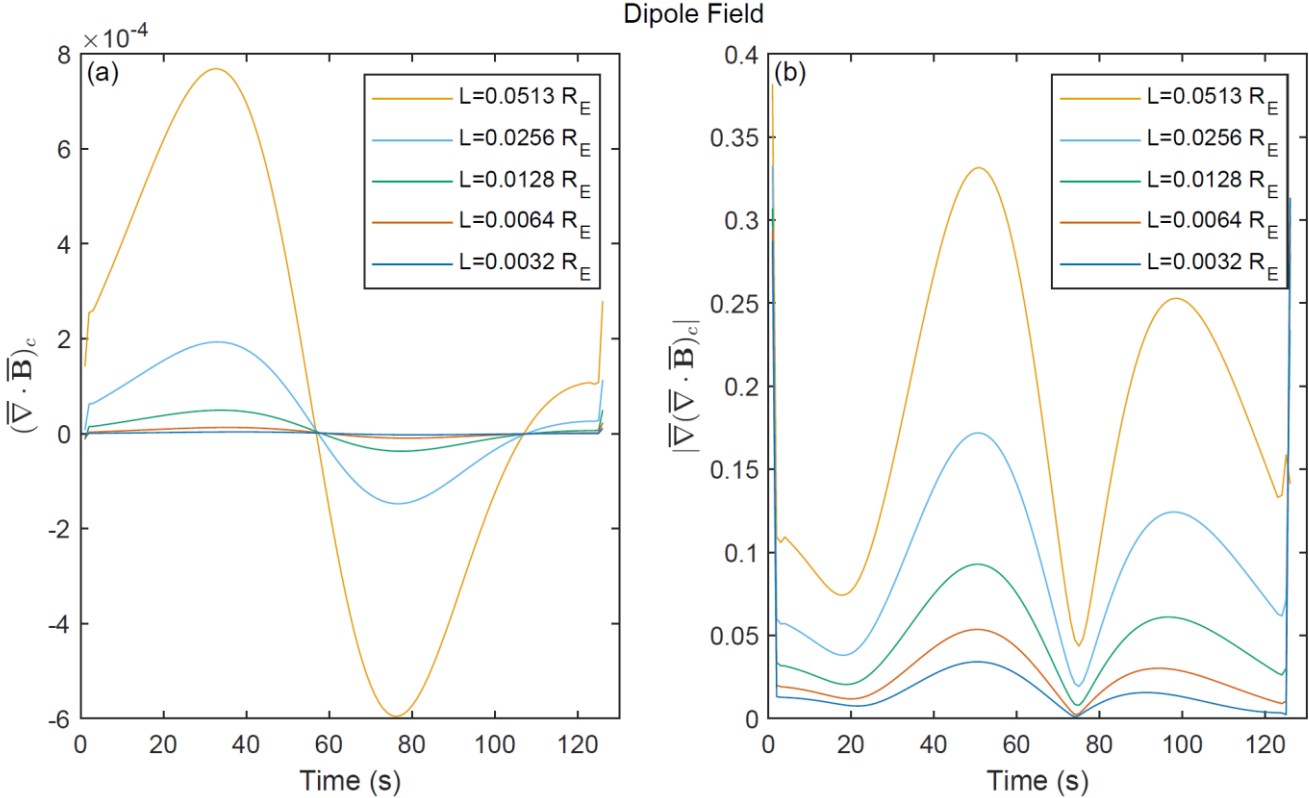

**Figure 13. Dimensionless divergence and gradient of divergence for magnetic field along the dipole field crossing path with different characteristic S/C sizes (L).**

The algorithm gradients were utilized to calculate the dimensionless divergence and the gradient of divergence for magnetic field at the barycentre along the crossing path with different characteristic S/C sizes, with the results for the flux-rope and dipole-field cases shown in Figure 12 and 13, respectively. Figure 12 (a) and 13 (a) show that the dimensionless divergence $\left(\overline{\nabla} \cdot \overline{\boldsymbol{B}}\right)_c$ at the barycentre is in the order of $10^{-4}$, while L varies from 0.0032 to 0.0513 $R_E$. The dimensionless gradient of the divergence $\left|\overline{\nabla}\left(\overline{\nabla} \cdot \overline{\boldsymbol{B}}\right)_c\right|$ for the flux-rope case was less than 0.02 with L=0.0513 $R_E$, as shown in Figure 12 (b). Similarly, Figure 13 (b) shows that $\left|\overline{\nabla}\left(\overline{\nabla} \cdot \overline{\boldsymbol{B}}\right)_c\right|$ was less than 0.4, with L=0.0513 $R_E$, for the dipole field. Meanwhile, that $\left|\overline{\nabla}\left(\overline{\nabla} \cdot \overline{\boldsymbol{B}}\right)_c\right|$ decreased with decreasing L in both cases. These results confirm the accuracy of the proposed method. As evidenced in Figures 12 and 13, the errors of the first derivative decrease quadratically with the scale L whereas the errors of the second derivatives decrease linearly with L.

## 5. Conclusions

In this study, a new algorithm was derived to estimate the linear and quadratic spatial gradients of the magnetic field from simultaneous 7- or 9-point magnetic measurements to obtain the fine structure of the magnetic field and the magnetic field geometry, allowing elucidation of whether the 7-spacecraft Plasma Observatory and the 9-spacecraft HelioSwarm missions could be utilized for such measurements. By inputting simultaneous 7–9-point magnetic measurements and using the reference frame transformation relationships of the magnetic field as well as the least-squares method, the new algorithm performs several iterations to finally derive the convergent magnetic linear and quadratic spatial gradients.

The developed algorithm requires only one restriction on the spatial configuration of the constellations, which is that the constellations must be non-planar. Actual operating constellations can easily satisfy this constraint. Only simultaneous magnetic measurements are required, with no other physical measurements needed, and the only physical constraint of the algorithm is the reference-frame transformation relationship of the magnetic field. In this study, simultaneous magnetic measurements from 7 or 9 points were assumed to be obtained by identical instruments onboard mothercraft and daughtercraft of the space mission. We also note that the magnetic field data from different detectors need to be synchronized by the time interpolation. In reality, a homogeneous set of instruments onboard spacecraft may not be achieved, and the temporal measurements at different detectors may not be perfectly synchronized. In this study, total truncation error has been evaluated. Furthermore, the iteration error, discretisation error and measurement error have been initially evaluated in Appendix. It is found that the iteration error, discretisation error, and measurement error are expected to be rather small for the dipole field case. Divergence-free magnetic field conditions were not required to calculate the magnetic spatial gradient. Alternatively, in the algorithm, the magnitudes of the magnetic divergence and its gradient were used to evaluate the truncation errors of the linear and quadratic magnetic spatial gradients, respectively.

The proposed algorithm was demonstrated using a cylindrical force-free flux rope and a dipole magnetic field, with results showing that the iterations effectively converged and that the magnetic spatial gradients can reach reasonable accuracy. The results of this study can thus be applied to the analysis of magnetic field data from multi-spacecraft constellations (e.g., the Plasma Observatory and HelioSwarm) as well as to the design of future constellation missions.

## Author contributions

CS designed the method and RK gives some advice. GZ developed the model code and performed the test. CS and GZ wrote the paper. CS and RK reviewed and edited the paper. YZ helps to evaluate the method.

**Competing interests**

The authors declare that they have no conflict of interest.

**Acknowledgments**

This work was supported by the National Natural Science Foundation of China (Grant No. 42130202),National Key Research and Development Program of China (Grant No. 2022YFA1604600) and Hubei Provincial Natural Science Foundation of China
(Grant No. 2022CFB928). This research was also supported by the International Space Science Institute (ISSI) in Bern through the ISSI International Team project #556 (cross-scale energy transfer in space plasmas). Work at IRAP is supported by CNRS, CNES, and UPS. We appreciate the reviewer's valuable suggestions.

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

# Appendix A: Iteration error

In the following error analysis, only the dipole field case is taken as an example. In section 3.2, the cutoff number of iterations was set to 100. In order to clearly show the convergence of the iterations, only the number of iterations was increased to 1000, while all other conditions are the same as section 3.2. As shown in Figure A1, the iterations are convergent.

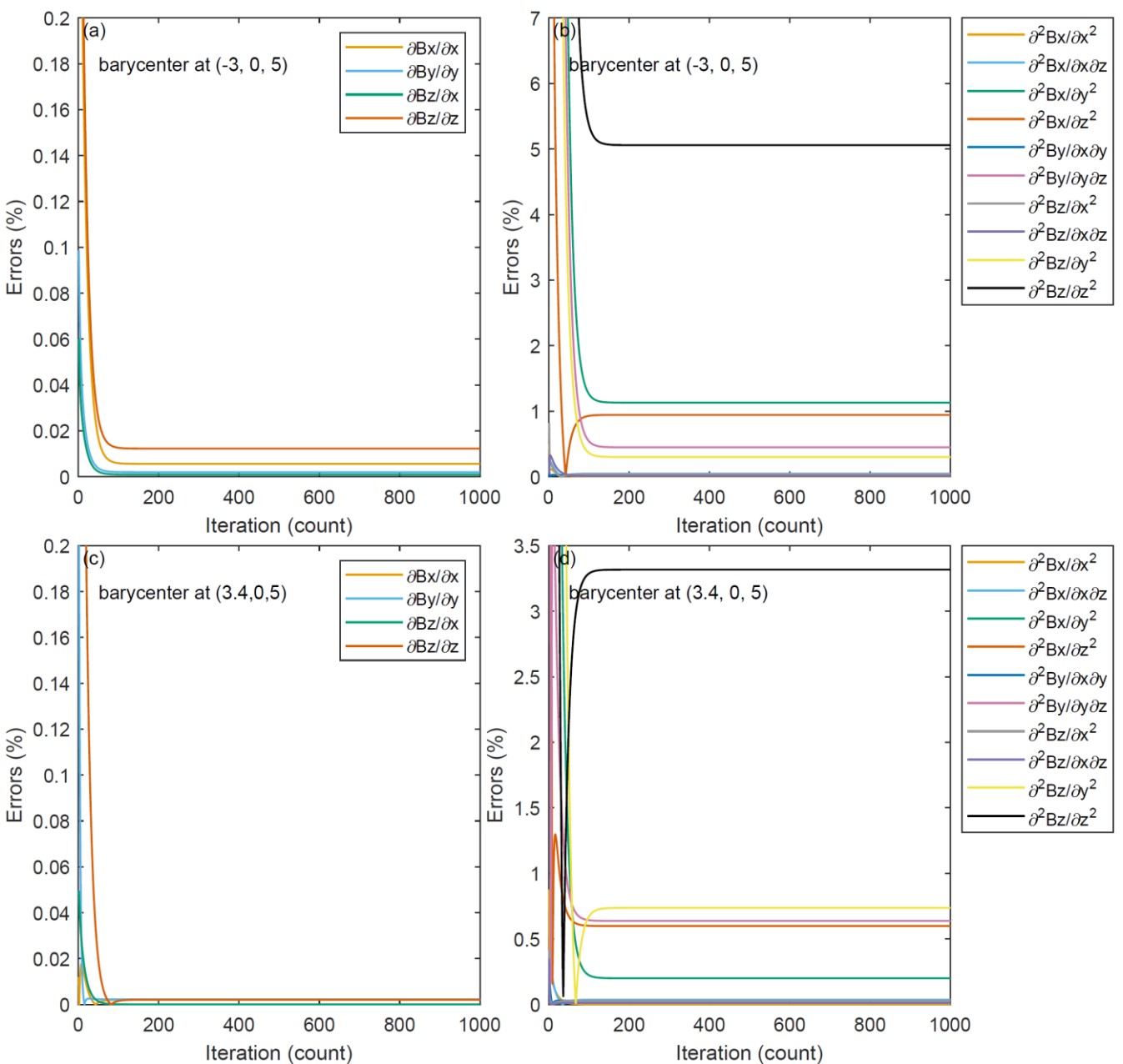

**Figure A1. Relative errors in the nonzero components of the linear and quadratic gradients with different iteration numbers at various barycentres for the dipole field case.**

By holding the configuration of the 7-S/C constellation, the distances between satellites are scaled down by a factor of 100, which decreased the characteristic size of the 7-S/C array to $L=0.2564\times10^{-3}$ $R_E$. Due to this reduction, the high-order truncation error converges to zero, leaving only the iteration error. Figure A2 shows the variation in relative errors of the linear

and quadratic gradients with respect to the iteration numbers for the dipole field case. The relative errors of the 4 nonzero components of linear gradients at point (-3, 0, 5) and (3.4, 0, 5) are both less than $10^{-6}$%. The relative errors of the 10 nonzero components of quadratic gradients at point (-3, 0, 5) are 0.0085%, 0.0459%, 0.0377%, 0.0928%, 0.0276%, 0.0701%, 0.0459%, 0.0083%, 0.0338% and 0.2880%, while those at point (3.4, 0, 5) are 0.0063%, 0.0304%, 0.0171%, 0.0870%, 0.0271%, 0.0788%, 0.0304%, 0.0180%, 0.0112% and 0.2405%, respectively. It is found that the relative errors of the linear gradients

are decreased to 0 and those of the majority quadratic gradients are decreased to less than 0.1%, with the distances between satellites reduced. It can be suggested that the error generated during the iteration process is relatively small.

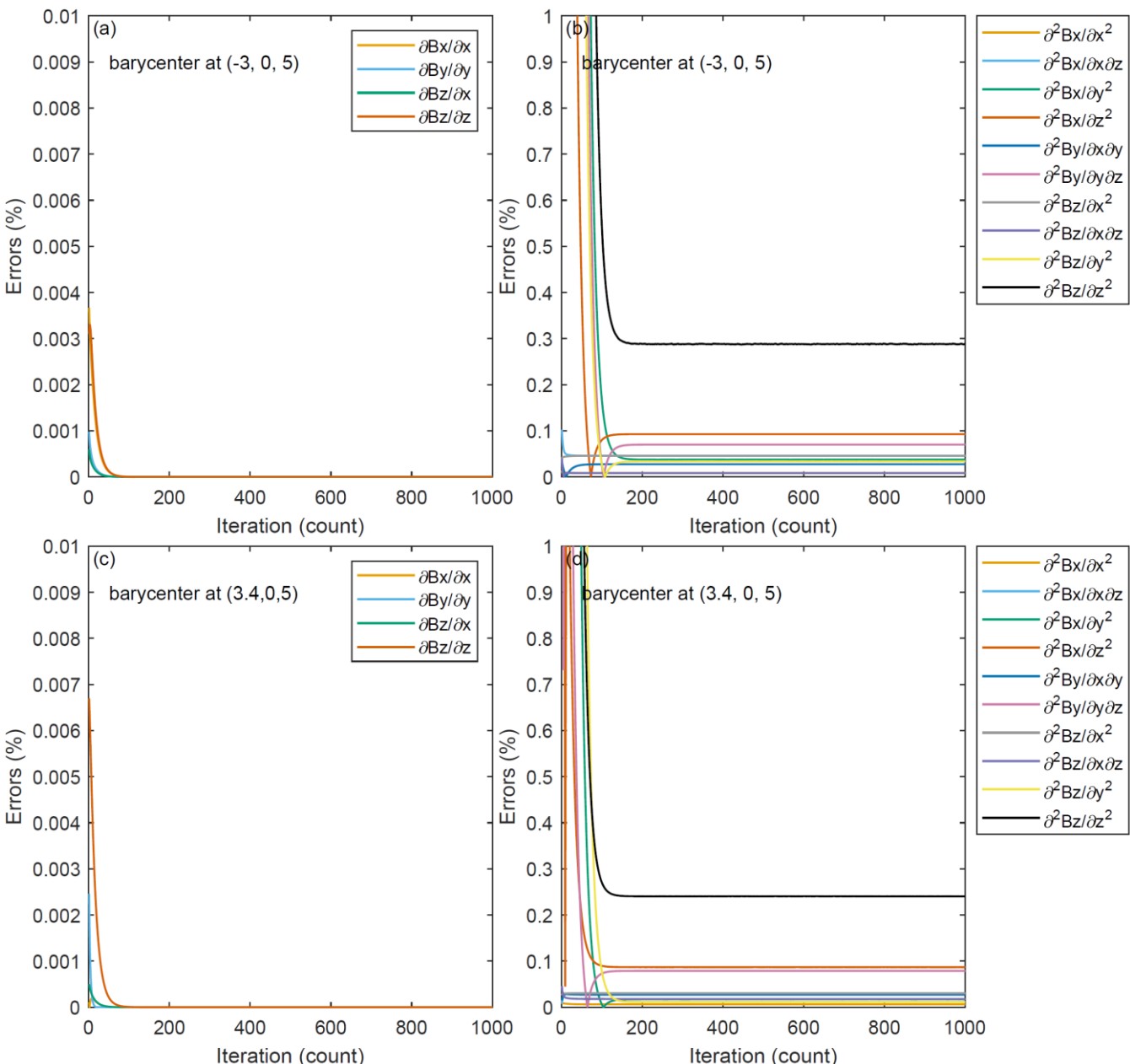

**Figure A2. As the truncation error converges to zero, relative errors in the nonzero components of the linear and quadratic gradients with different iteration numbers at various barycentres for the dipole field case.**

**Appendix B: Discretisation error**

It is assumed that the magnetic field value at the measurement point is the average along the satellite's trajectory for a duration of 0.25 seconds before and after the point, in the direction of the satellite's motion. This assumption introduces a discretisation error. The characteristic size of the 7-S/C array is decreased to $L=0.2564\times10^{-3}$ $R_E$. So, the truncation error converges to zero, leaving only the discretisation error. Figure B1 shows the variation in relative errors of the linear and quadratic gradients with respect to the iteration numbers for the dipole field case, with a discretisation error introduced. The relative errors of the 4 nonzero components of linear gradients at point (-3, 0, 5) are 0.0122%, $8.3688\times10^{-4}$%, $7.4499\times10^{-4}$% and 0.0066%, while those at point (3.4, 0, 5) are 0.0062%, 0.0011%, $1.8504\times10^{-5}$% and 0.0113%, respectively. The relative errors of the 10 nonzero components of quadratic gradients at point (-3, 0, 5) are 0.0063%, 0.0524%, 0.0387%, 0.0936%, 0.0268%, 0.0724%, 0.0524%, 0.0091%, 0.0359% and 0.1082%, while those at point (3.4, 0, 5) are 0.0121%, 0.0345%, 0.0170%, 0.0845%, 0.0269%, 0.0759%, 0.0344%, 0.0204%, 0.0141% and 0.2594%, respectively. The relative errors of linear gradients are less than 0.012%, while those of majority quadratic gradients are less than 0.1%. It can be suggested that the discretisation error is relatively small.

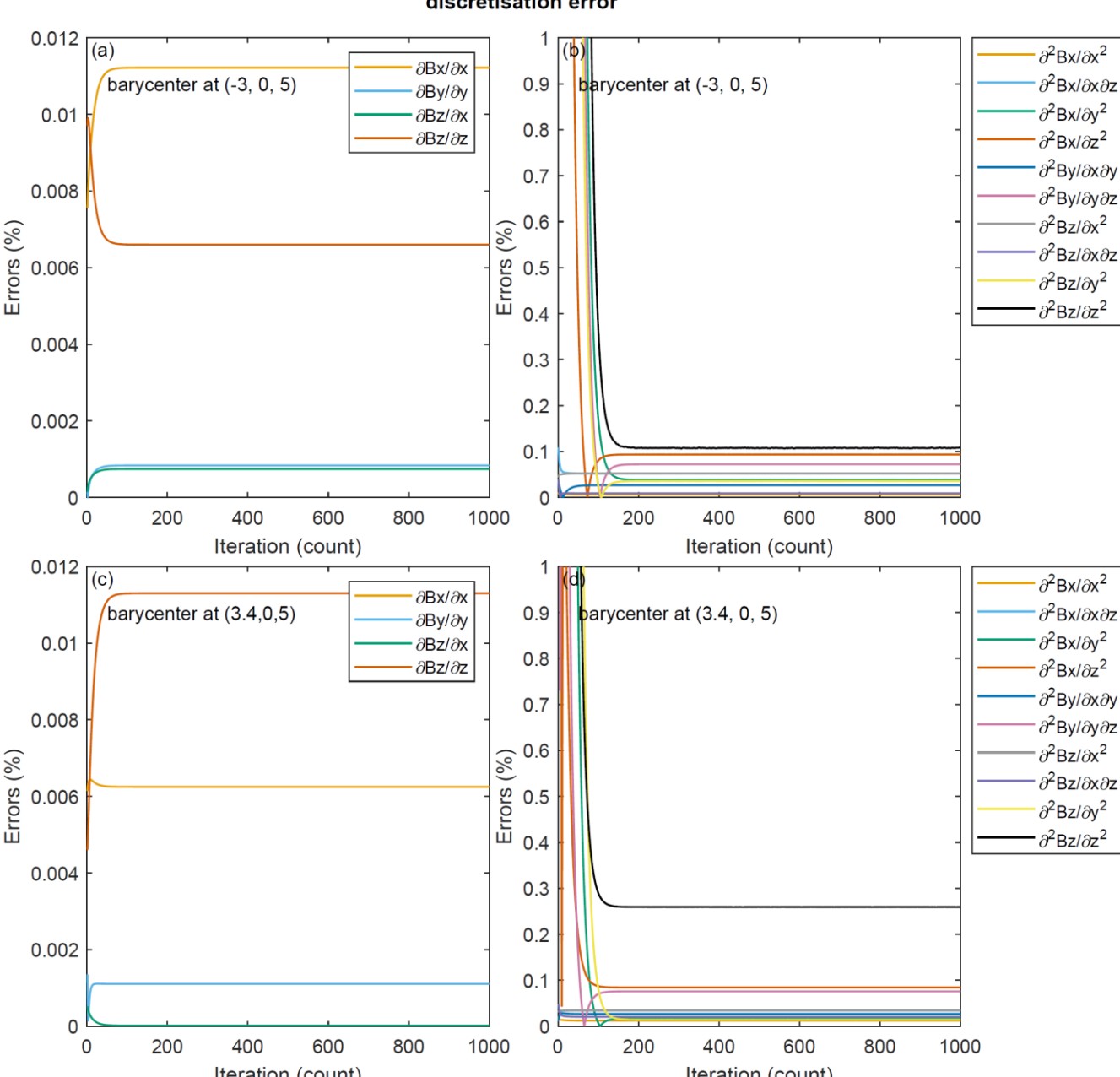

**Figure B1. As the truncation error converges to zero, relative errors of the linear and quadratic gradients with different iteration numbers at various barycentres for the dipole field case, with a discretisation error introduced.**

**Appendix C: Measurement error**

In order to evaluate measurement error, a measurement error of 0.1% is introduced, i.e. each measurement is decreased by 0.1%. The characteristic size of the 7-S/C array is decreased to $L=0.2564\times10^{-3}$ $R_E$, so the truncation error converges to zero. And there is no discretisation error, only measurement error remains. Figure C1 shows the variation in relative errors of the linear and quadratic gradients with respect to the iteration numbers for the dipole field case, with a measurement error of 0.1% introduced. The relative errors of the 4 nonzero components of linear gradients at point (-3, 0, 5) and (3.4, 0, 5) both are exactly 0.1%. The relative errors of the 10 nonzero components of quadratic gradients at point (-3, 0, 5) are 0.1085%, 0.1458%, 0.1377%, 0.0073%, 0.0724%, 0.0300%, 0.1459%, 0.0917%, 0.0663% and 0.1881%, while those at point (3.4, 0, 5) are 0.0937%, 0.1304%, 0.0829%, 0.1869%, 0.0729%, 0.1787%, 0.1303%, 0.0820%, 0.0889% and 0.3403%, respectively. The relative errors of the majority quadratic gradients are less than 0.2%. Compare with Figure A2, it is found that the relative errors of the gradients are increased about 0.1%, which is the same as the introduced measurement error of 0.1%.

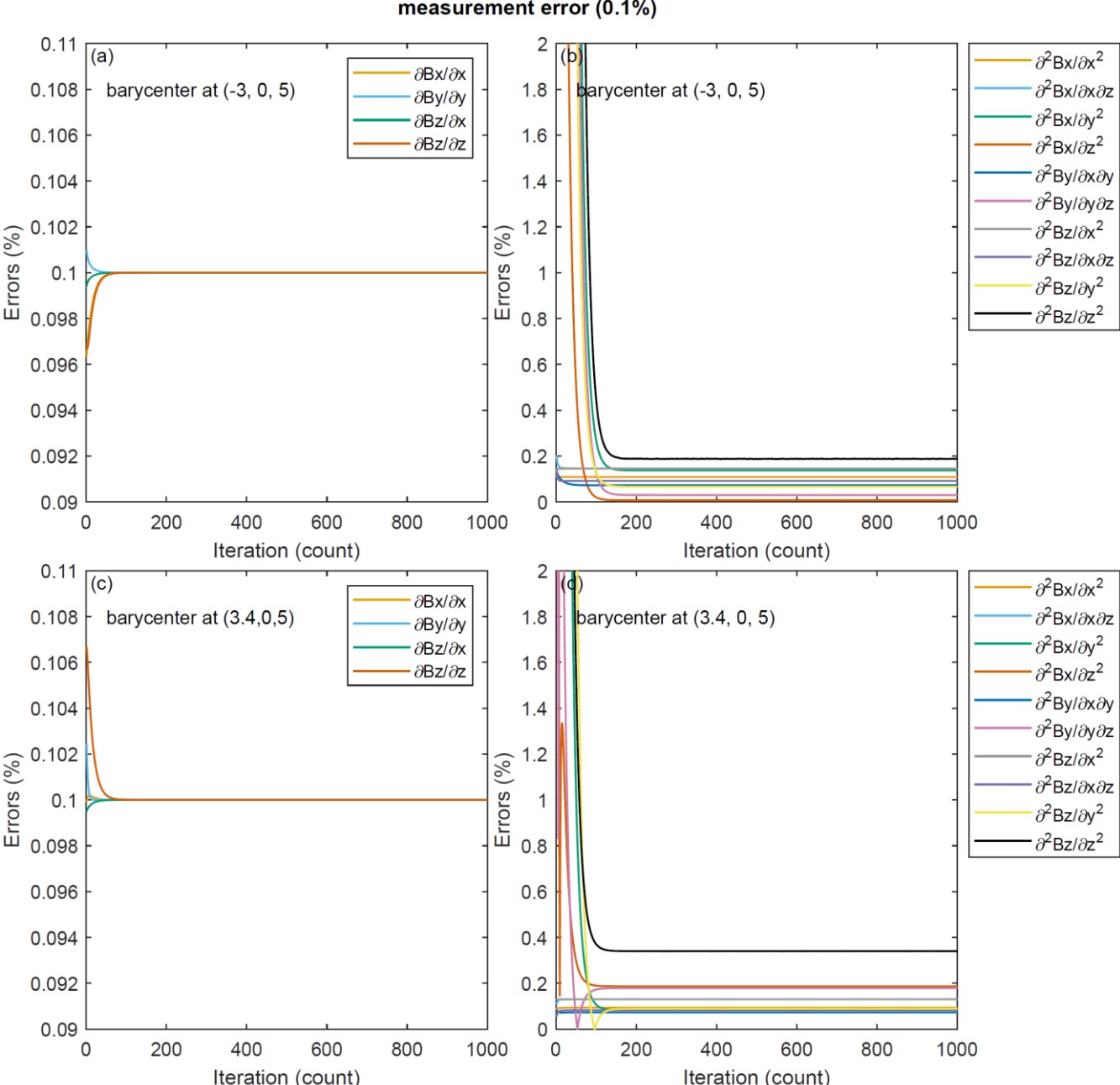

**Figure C1. As the truncation error converges to zero, relative errors of the linear and quadratic gradients with different iteration numbers at various barycentres for the dipole field case, with a measurement error of 0.1% introduced.**

A measurement error of 1% is introduced to re-evaluate the relative error. Figure C2 shows the variation in relative errors of the linear and quadratic gradients with respect to the iteration numbers for the dipole field case, with a measurement

error of 1% introduced. The relative errors of the 4 nonzero components of linear gradients at point (-3, 0, 5) and (3.4, 0, 5)

both are exactly 1%. The relative errors of the 10 nonzero components of quadratic gradients at point (-3, 0, 5) are 1.0085%,

1.0454%, 1.0374%, 0.9081%, 0.9727%, 0.9306%, 1.0454%, 0.9918%, 0.9667% and 0.7154%, while those at point (3.4, 0, 5)

are 0.9938%, 1.0301%, 0.9831%, 1.0861%, 0.9732%, 1.0780%, 1.0301%, 0.9822%, 0.9890% and 1.2383%, respectively. The

relative errors of the quadratic gradients are around 1%. Compare with Figure A2, it is found that the relative errors of the

gradients are increased about 1%, which is also the same as the introduced measurement error of 1%. It can be suggested that

the measurement error is of the same order as the accuracy of the instrument.

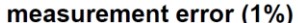

**Figure C2. As the truncation error converges to zero, relative errors of the linear and quadratic gradients with different iteration numbers at various barycentres for the dipole field case, with a measurement error of 1% introduced.**

The relative errors of the linear and quadratic gradients, with different types of errors at various barycentres, are summarized in Tables C1 and C2 for the dipole field case.

**Table C1. relative errors of the linear gradients with different errors at various barycentres for the dipole field case**

| relative error | | $\partial B_x/\partial x$ | $\partial B_y/\partial y$ | $\partial B_z/\partial x$ | $\partial B_z/\partial z$ |
|---|---|---|---|---|---|
| point (-3, 0, 5) | truncation error converges to 0 | $2.5036\times10^{-6}$% | $6.8185\times10^{-7}$% | $6.2787\times10^{-7}$% | $1.9854\times10^{-6}$% |
| | discretisation error | 0.0122% | $8.3688\times10^{-4}$% | $7.4499\times10^{-4}$% | 0.0066% |
| | measurement error (0.1%) | 0.1% | 0.1% | 0.1% | 0.1% |
| | measurement error (1%) | 1% | 1% | 1% | 1% |
| point (3.4, 0, 5) | truncation error converges to 0 | $1.0170\times10^{-6}$% | $6.9004\times10^{-7}$% | $3.6922\times10^{-7}$% | $1.8656\times10^{-6}$% |
| | discretisation error | 0.0062% | 0.0011% | $1.8504\times10^{-5}$% | 0.0113% |
| | measurement error (0.1%) | 0.1% | 0.1% | 0.1% | 0.1% |
| | measurement error (1%) | 1% | 1% | 1% | 1% |

**Table C2. relative errors of the quadratic gradients with different errors at various barycentres for the dipole field case**

| relative error | | $\partial^2 B_x/\partial x^2$ | $\partial^2 B_x/\partial x\partial z$ | $\partial^2 B_x/\partial y^2$ | $\partial^2 B_x/\partial z^2$ | $\partial^2 B_y/\partial x\partial y$ | $\partial^2 B_y/\partial y\partial z$ | $\partial^2 B_z/\partial x^2$ | $\partial^2 B_z/\partial x\partial z$ | $\partial^2 B_z/\partial y^2$ | $\partial^2 B_z/\partial z^2$ |
|---|---|---|---|---|---|---|---|---|---|---|---|
| point (-3, 0, 5) | truncation error converges to 0 | 0.0085% | 0.0459% | 0.0377% | 0.0928% | 0.0276% | 0.0701% | 0.0459% | 0.0083% | 0.0338% | 0.2880% |
| | discretisation error | 0.0063% | 0.0524% | 0.0387% | 0.0936% | 0.0268% | 0.0724% | 0.0524% | 0.0091% | 0.0359% | 0.1082% |
| | measurement error (0.1%) | 0.1085% | 0.1458% | 0.1377% | 0.0073% | 0.0724% | 0.0300% | 0.1459% | 0.0917% | 0.0663% | 0.1881% |
| | measurement error (1%) | 1.0085% | 1.0454% | 1.0374% | 0.9081% | 0.9727% | 0.9306% | 1.0454% | 0.9918% | 0.9667% | 0.7154% |
| point (3.4, 0, 5) | truncation error converges to 0 | 0.0063% | 0.0304% | 0.0171% | 0.0870% | 0.0271% | 0.0788% | 0.0304% | 0.0180% | 0.0112% | 0.2405% |
| | discretisation error | 0.0121% | 0.0345% | 0.0170% | 0.0845% | 0.0269% | 0.0759% | 0.0344% | 0.0204% | 0.0141% | 0.2594% |

| | measurement error (0.1%) | 0.0937% | 0.1304% | 0.0829% | 0.1869% | 0.0729% | 0.1787% | 0.1303% | 0.0820% | 0.0889% | 0.3403% |
|---|---|---|---|---|---|---|---|---|---|---|---|
| | measurement error (1%) | 0.9938% | 1.0301% | 0.9831% | 1.0861% | 0.9732% | 1.0780% | 1.0301% | 0.9822% | 0.9890% | 1.2383% |