# Peer review of "Quadratic Magnetic Gradients from 7- and 9-Spacecraft Constellations"

_EGUsphere, 2024_

## Author Comment (AC1)

**Replies to the first Reviewer**

We are very grateful for the referee's valuable comments. We address your suggestions point-by-point below and improve our manuscript presentation throughout. Major changes include the structuring of the introduction and adding more context to make the manuscript accessible to a broader group of audience. We hope that our revised manuscript is now more accessible to the general readers. In this reply, the comments of the referee are marked in black or red colors, and the replies in blue color.

Review of egusphere-2024-1330 (ANGEO)

Quadratic Magnetic Gradients from 7-SC and 9-SC Constellations

by Chao Shen et al.

This paper describes a least-squares gradient computation technique for linear and quadratic magnetic gradients. The technique is applied to two test cases to show its performance. One of the goals is to demonstrate that 7- and 9-spacecraft constellations provide enough measurements to infer those gradients. The paper starts with an introduction that properly references earlier work on gradient computation. It then presents the technique, the test cases, and it ends with a conclusion.

The introduction could be better structured. This can probably be remedied by shifting some material from the description of the technique to the introduction, so that the characteristics of the technique are put in contrast with the earlier work on the subject (see details below). The actual contents of the paper is sound and will undoubtedly be useful for the community. I do have a number of questions/suggestions regarding the method, the test cases, and the presentation (see comments below).

The manuscript would benefit seriously from language editing. I have listed just a few language suggestions (see below).

**Major comments**

In the abstract and at various places in the text, the authors say that 4 measurements are needed for computing the linear gradient and 10 measurements are needed for the

quadratic gradient. This statement is somewhat imprecise. It would be more correct to state instead that 4 simultaneous measurements are needed for the linear spatial gradient and 10 simultaneous measurements for the quadratic spatial gradient components of a scalar field. Perhaps it would also be useful to mention from the start that, when using the least-squares approach, one adds the time derivatives and the mixed space-time derivatives, so that at least 5 measurements are needed for the linear and 15 for the non-linear gradients of a scalar field in general.

Reply:

Thank you for these valuable comments. To address your points, we have made the following modifications.

1. We have now emphasized that we require multi-point 'simultaneous' measurements everywhere in the text.

2. We have emphasized that 4-point simultaneous measurements are needed to resolve the linear spatial gradient of a scalar field in Paragraph 1 of the introduction.

3. We have elaborated the necessity of 10-point simultaneous measurements to resolve the quadratic spatial gradient of a magnetic field in Paragraph 3 of the introduction. Furthermore, we have further generalized potential applications of quadratic spatial gradients, in addition to resolving complex magnetic structures such as flux ropes, to include nonlinear plasma dynamics that would benefit also from the high-order magnetic gradient calculation from measurements *in situ* at the end of the same paragraph.

Generally, the number of the measurement points required for drawing till the r-th gradients in the d dimensional space is $C_{d+r}^r = \dfrac{(d+r)!}{d!\,r!}$ ( Zhou & Shen, 2024). For the situation considered in this study (to obtain the 1st and 2nd magnetic gradients in 3 dimensional space), r=2, d=3. Thus the spacecraft needed in the constellation is at least $C_{d+r}^r = \dfrac{(3+2)!}{3!\,2!} = 10$.

Ref: Zhou, Y. and Shen, C.: Estimating gradients of physical fields in space, Ann. Geophys., 42, 17–28. https://doi.org/10.5194/angeo-42-17-2024, 2024.

4. We have added that, to compute quadratic gradients from 7- or 9-point simultaneous measurements, we consider the transformation of reference frame involving mixed space-time derivatives of the magnetic field at the end of Paragraph 4 of the introduction. We also specified in the methodology that we consider the mixed space-time derivatives to avoid confusion.

In the description of the method, I was expecting that somewhere the condition div B = 0 would have been incorporated. If I understand well, that is not the case; rather that condition is used for evaluating the precision of the technique. Still, inclusion of a div B = 0 constraint would make the technique more precise and robust, as it can remove a possible ill-posedness of the problem for certain spacecraft constellation geometries. Can the authors comment on whether and how such a condition can be included?

Reply:

Here, the transformation relationship constraints are sufficient already for obtaining the complete linear spatial gradient and quadratic spatial magnetic gradient.

Certainly, applying the div B = 0 and grad div B=0 constraints can improve the algorethim, but not very significantly. And also, grad div B=0 can only provide two constraints equations (the gradient of div B along the motion direction can be obtained from the transformation relationship).

Alternatively, we only apply the transformation relationship constraints in this method, While the div B = 0 and grad div B=0 constraints are used as the quantitative measures of the errors of the magnetic gradient calculated in this algorithm. By the way, the Curlometer technique (Dunlop et al., 2002b) to calculate the current density based on multiple spacecraft magnetic measurements has used abs (div B/curl B) to evaluate the error.

Nevertheless, the previous method (NMG, Shen et al., 2021a) for calculating the linear and quadratic magnetic gradients based on 4-spacecraft MMS mission observations has to apply both the div B = 0 and grad div B=0 constraints. Several other methods also utilize the div B = 0 or both the div B = 0 grad div B=0 constraints (Liu et al., 2019; Torbert et al., 2020).

Ref:

Dunlop, M. W., Balogh, A., Glassmeier, K.-H., and Robert, P.: Four-point cluster application of magnetic field analysis tools: The curlometer, J. Geophys. Res., 107, 1384. https://doi.org/10.1029/2001JA005088, 2002b.

Shen, C., Zhang, C., Rong, Z., Pu, Z., Dunlop, M. W., Escoubet, C. P., Russell, C. T., Zeng, G., Ren, N., Burch, J. L., Zhou, Y.: Nonlinear magnetic gradients and complete magnetic geometry from multispacecraft measurements, J. Geophys. Res., 126, JA028846. https://doi.org/10.1029/2020JA028846, 2021a.

For the reader it is confusing that the time derivative is used (line 109) in the explanation of the technique, while time derivatives or mixed space-time derivatives do not appear in the variable count on lines 122ff.

Reply:

Here we use the time derivatives and mixed space-time derivatives to add more constraints in order to obtain a unique solution of quadratic gradients from 7- or 9-point observations. We agree that this can lead to confusion as our work is focused on estimation of spatial gradients only. To avoid the confusion, we now specified everywhere whether it is a 'spatial' or 'temporal' gradient where applicable. Also, in our previous version of the manuscript, our assumption involving the temporal change of the magnetic structure was not clearly stated. To better clarify this point, we now emphasized that we assume that the magnetic structures (e.g., flux ropes, current sheets, boundary layers, magnetic reconnection regions, etc.) are slowly evolving during their passages through the multi-point constellations. This assumption has now explicitly been stated in Paragraph 1 of Section 2.

For actual magnetic observations, the time resolution is very high (~0.01sec), there are plenty of time series data, so it is not difficult to get the time derivative of magnetic field even in high orders. We are concentrated on the spatial gradients of magnetic field in this study.

I think having the paragraph from line 122ff in the introductory section would help in setting the broader problem of balancing the number of unknowns versus the number of available observations.

Reply:

The introduction on the system of equations was introduced briefly in the 2$^{nd}$ paragraph of our original manuscript. To better introduce the setting of our problem, we expand the introduction to include these.

The discussion of the volume tensor states that its determinant should be nonzero. At this point, no mention is made of the condition number, which is – practically speaking – more important than the tensor being non-singular. The statement that "This algorithm requires that the constellation be composed of at least seven spacecraft and that its configuration is non-planar. Because both the 9S/C HelioSwarm and 7S/C Plasma Observatory satisfy these requirements, the linear and quadratic magnetic gradients can be readily obtained" is therefore perhaps a bit optimistic. It is appreciated that in the

examples the eigenvalues of the volume tensor are given. Still, that only partially describes the conditioning of the problem.

Reply:

Thanks for the comments. For the random shape of the 7S/C in the tests, the three eigenvalues of the volumetric tensor are $w_1 = 0.1643 \times 10^{-3} R_E^2$, $w_2 = 0.1104 \times 10^{-3} R_E^2$, and $w_3 = 0.0341 \times 10^{-3} R_E^2$. And it is already written in this new version of the manuscript.

It is certain that we have not verified the conditioning mentioned. However, it is strictly correct. From the equation (12) in the subsection 2.2.1, we have speculated that "The constellation must be nonplanar to achieve this result" without a verification. However, this could be made clear based on the results obtained in the previous work on high order gradients of physical fields this year (Zhou & Shen, 2024). The equation (12) is that for the parameters $G_{rs}^{(1)}(r,s=1,2)$, i.e., $\left(G_{11}^{(1)}, G_{12}^{(1)}, G_{21}^{(1)}, G_{22}^{(1)}\right)$. Similar to the analysis in the subsection 4.1 of Zhou & Shen (2024), it is expected that, in order the solution exists the position of all the spacecraft in the contellation must not obey the following formula

$$a_{11}x_1^2 + a_{12}x_1x_2 + a_{12}x_2x_1 + a_{22}x_2^2 = 0, \tag{1}$$

Where $a_{rs}$ is fixed coefficients. The above equation can be rewritten as

$$a_{11}(x_1/x_2)^2 + 2a_{12}(x_1/x_2) + a_{22} = 0, \tag{2}$$

Which means that all the spacecraft are in the plane parallel to the x3 axis or the motion direction. Therefore it is necessary that the constellation should not be planar in order to deduce the quadratic magnetic gradients as well as the linear magnetic gradient. The next iterations would also require this condition. So this would verify the statement.

We need to make an explanation in the revised manuscript accordingly.

Ref: Zhou, Y. and Shen, C.: Estimating gradients of physical fields in space, Ann. Geophys., 42, 17–28. https://doi.org/10.5194/angeo-42-17-2024, 2024.

Figure 1 presents a very specific shape of the 7 S/C constellation. Such a constellation is nice for conceptually presenting the idea of "nested tetrahedra", but cannot be easily maintained in space in practice. This figure is nowhere referenced nor discussed.

Reply:

Figure 1 just presents a schematic diagram of the Plasma Observatory Constellation. It is not the actual shape of the constellation. In the tests in section 3, the positions of the seven spacecrafts are generated randomly and is illustrated in Figure 2. The shape of the constellation in the tests is not the same as that in Figure 1. It is noted in the caption of Figure 1 specially.

The effect of measurement errors is not included in the calculation. This is assuming a homogeneous set of instruments, but that may not be the case for Plasma Observatory, for instance, where there are different instruments on mother and daughter spacecraft.

Reply:

In this study, the homogeneous measurement has been adopted in both the method and tests. For actual multi-satellite measurements, the measurement could be not homogeneous. The magnetic field data from different detectors would be synchronized by time interpolation. This is explained in the Conclusions section.

As for the measurement error, the influence of the magnetic detector on the output of method is a complicated problem. In the previous manuscript, we only checked the errors originated from the method itself or the truncation errors.

Regarding the error caused by the measurements, we may make an initial estimation. Starting from the formulas (1) and (2) in the Section 2, we have

$$\Delta f_{(\alpha)} = f_{(\alpha)} - f_c = x^i_{(\alpha)} g_i + \frac{1}{2} x^i_{(\alpha)} x^j_{(\alpha)} G_{ij} \tag{3}$$

Considering the error of the measurement on the positions of the satellite are generally very small and can be neglected, the above equations are linear. It can be expected that the relative errors of gradients are roughly estimated as

$$\frac{\delta g_i}{|g_i|} \sim \frac{\delta f}{|\Delta f|}, \quad \frac{\delta G_{ij}}{|G_{ij}|} \sim \frac{\delta f}{|\Delta f|} \tag{4}$$

Assuming the typical magnetic strength of structure is B, the characteristic

spatial scale of the structure is D, while the separation of the satellite is L, then

$$|\Delta f| \sim \frac{LB}{D} \tag{5}$$

So that the relative error is about

$$\frac{\delta f}{|\Delta f|} \sim \frac{\delta B}{LB/D} = \frac{\delta B}{B} \frac{D}{L} \tag{6}$$

E.g., for the magnetotail measurements, B~20nT, D~2000km, L~200km, $\delta B$ ~0.01nT. Then the relative error is

$$\frac{\delta B}{B} \frac{D}{L} \approx \frac{0.01 nT}{20 nT} \cdot \frac{2000 km}{200 km} \sim 0.005 \tag{7}$$

Therefore, the method may be valid.

Presently, the instruments of HelioSwarm and Plasma Observatory constellations are still not fixed and their errors are not available. So we think that it is proper to make a restrict evaluation on the measurement errors in the future and after the operations of the missions.

Nothing is said about error estimates on the results (in the case where you do not know the exact solution). Does the technique allow you to produce such error estimates? If so, it would be useful to compare these estimates to the actual errors for the two test cases.

Reply:

The error is evaluated in section 4. The divergence and gradient of divergence obtained from algorithm are used to evaluate the error. To offer a uniform standard for evaluation, the divergence and gradient of divergence were non-dimensionalized with the corresponding characteristic quantity.

Minor comments

-    Title: Personally, I would try to avoid the "SC" abbreviation in the title. Better change into: "Quadratic Magnetic Gradients from 7- and 9-Spacecraft Constellations"
Reply:

Done. Thanks.

The abbreviation has been deleted in the title and been substituted by full writing accordingly.

line 12: remove "therefore"

Reply:

The correction has been made accordingly.

line 13: from -> from the

Reply:

The correction has been made accordingly.

line 17: The tests -> Tests

Reply:

The correction has been made accordingly.

line 18: verifies -> verified

Reply:

The correction has been made accordingly.

line 23: iteration algorithm -> iterative algorithm

Reply:

The correction has been made accordingly.

line 38: gradient -> gradients

Reply:

The correction has been made accordingly.

line 43: tetrahedral -> a tetrahedral

Reply:

The correction has been made accordingly.

line 43: such the missions -> such missions

Reply:

The correction has been made accordingly.

line 62: consisting -> consisting of

Reply:

The correction has been made accordingly.

line 66: an ESA's new mission -> a new ESA mission

Reply:

The correction has been made accordingly.

line 68: drawn -> inferred

Reply:

The correction has been made accordingly.

line 74ff: I suggest to change punctuation into: "a description of the tests conducted for two typical magnetic structures (a cylindrical force-free flux rope and a dipole magnetic field), which were utilized to check the validity and accuracy of the new algorithm, is given …"

Reply:

Punctuations have been changed accordingly.

line 76: error -> accuracy

Reply:

The correction has been made accordingly.

line 84: references -> reference frames

Reply:

The correction has been made accordingly.

line 84: … of the magnetic field

Reply:

The correction has been made accordingly.

caption of Figure 1: relative to the constellations -> relative to the constellation

Reply:

The correction has been made accordingly.

line 95, 97, 195 and elsewhere: no capital needed at the beginning of the line

Reply:

Corrections have been made accordingly.

line 99: draw -> infer

Reply:

The correction has been made accordingly.

line 225: "The characteristic size of the S/C is twice the square root of the maximum eigenvalue" makes no sense. Size of the S/C constellation?
Reply:

Here we follow the definition of Harvey (1998) on the "characteristic size" in the tool book (Section 12.4.3). The characteristic size is used to indicate the size of the polyhedron and is twice the square root of the maximum eigenvalue of the volumetric tensor.

Ref: Harvey, C. C.: Spatial gradients and the volumetric tensor, in: Analysis methods for multi-spacecraft data, edited by Paschmann, G. and Daly, P. W., European Space Agency Publ. Division, Noordwijk, Netherlands, 315-319, 1998.

Fig 4, 6, 8: light yellow lines are hardly visible
Reply: Color of lines in the figures have been changed accordingly.

explain abbreviations when first used: NASA, ESA

Reply:

Abbreviations have been explained accordingly.

---

## Author Response (AR1)

**Replies to the first Reviewer**

Dear Editor,

We are very grateful for the referee's valuable comments. We address your suggestions point-by-point below and improve our manuscript presentation throughout. Major changes include the structuring of the introduction and adding more context to make the manuscript accessible to a broader group of audience.

Since we solicited a significant amount of expertise from a colleague to improve this manuscript, we have added an additional co-author to this revised version of the manuscript. In this reply, the comments of the referee are marked in black and the replies in blue. We hope that our revised manuscript is now more accessible to the general readers.

Yours sincerely,

Chao Shen, on behalf of the co-authors.
* * *
**Review of egusphere-2024-1330 (ANGEO)**

Quadratic Magnetic Gradients from 7-SC and 9-SC Constellations

by Chao Shen et al.

This paper describes a least-squares gradient computation technique for linear and quadratic magnetic gradients. The technique is applied to two test cases to show its performance. One of the goals is to demonstrate that 7- and 9-spacecraft constellations provide enough measurements to infer those gradients. The paper starts with an introduction that properly references earlier work on gradient computation. It then presents the technique, the test cases, and it ends with a conclusion.

The introduction could be better structured. This can probably be remedied by shifting some material from the description of the technique to the introduction, so that the characteristics of the technique are put in contrast with the earlier work on the subject (see details below). The actual contents of the paper is sound and will undoubtedly be useful for the community. I do have a number of questions/suggestions regarding the method, the test cases, and the presentation (see comments below).

The manuscript would benefit seriously from language editing. I have listed just a few language suggestions (see below).

We thank the referee for these valuable comments. In this version, we have improved the structuring of the manuscript throughout especially in the introduction and at the

beginning of the methodology section in the manuscript. Please see our replies to your major comments point-by-point below.

**Major comments**

In the abstract and at various places in the text, the authors say that 4 measurements are needed for computing the linear gradient and 10 measurements are needed for the quadratic gradient. This statement is somewhat imprecise. It would be more correct to state instead that 4 simultaneous measurements are needed for the linear spatial gradient and 10 simultaneous measurements for the quadratic spatial gradient components of a scalar field. Perhaps it would also be useful to mention from the start that, when using the least-squares approach, one adds the time derivatives and the mixed space-time derivatives, so that at least 5 measurements are needed for the linear and 15 for the non-linear gradients of a scalar field in general.

Reply:

Thank you for these useful feedbacks. To address your points, we have made the following modifications.

1. We have now emphasized that we require multi-point 'simultaneous' measurements everywhere in the text.

2. We have emphasized that 4-point simultaneous measurements are needed to resolve the linear spatial gradient of a scalar field in Paragraph 1 of the introduction.

3. We have elaborated the necessity of 10-point simultaneous measurements to resolve the quadratic spatial gradient of a magnetic field in Paragraph 3 of the introduction. Furthermore, we have further generalized potential applications of quadratic spatial gradients, in addition to resolving complex magnetic structures such as flux ropes, to include nonlinear plasma dynamics that would benefit also from the high-order magnetic gradient calculation from measurements *in situ* at the end of the same paragraph.

Generally, the number of the measurement points required for drawing till the r-th gradients in the d dimensional space is $C_{d+r}^r = \dfrac{(d+r)!}{d!\,r!}$ ( Zhou & Shen, 2024). For the situation considered in this study (to obtain the 1st and 2nd magnetic gradients in 3-dimensional space), r=2, d=3. Thus, the number of spacecrafts needed in the constellation is at least $C_{d+r}^r = \dfrac{(3+2)!}{3!\,2!} = 10$.

Ref: Zhou, Y. and Shen, C.: Estimating gradients of physical fields in space, Ann. Geophys., 42, 17–28. https://doi.org/10.5194/angeo-42-17-2024, 2024.

4. We have added that, to compute quadratic gradients from 7- or 9-point simultaneous measurements, we consider the transformation of reference frame involving mixed space-time derivatives of the magnetic field at the end of Paragraph 4 of the introduction. We also specified in the methodology that we consider the mixed space-time derivatives to avoid confusion.

In the description of the method, I was expecting that somewhere the condition div B = 0 would have been incorporated. If I understand well, that is not the case; rather that condition is used for evaluating the precision of the technique. Still, inclusion of a div B = 0 constraint would make the technique more precise and robust, as it can remove a possible ill-posedness of the problem for certain spacecraft constellation geometries. Can the authors comment on whether and how such a condition can be included?

Reply:

Here, the transformation relationship constraints are already sufficient for obtaining the complete linear spatial gradient and quadratic spatial magnetic gradient.

Certainly, applying the div B = 0 and grad div B=0 constraints can improve the algorithm, but not very significantly in our case. Also, grad div B=0 can only provide two constraints equations (the gradient of div B along the motion direction can be obtained from the transformation relationship).

Alternatively, we only apply the transformation relationship constraints in this method. Nevertheless, we note that the div B = 0 and grad div B=0 constraints are used as the quantitative measures of the errors of the magnetic gradient calculated in this algorithm as mentioned at the end of Section 2.2.2 and previously described in Section 4.

We acknowledge that the div B = 0 constraint has commonly been used in spatial gradient estimation. For instant, the Curlometer technique (Dunlop et al., 2002b) for the calculation of current density based on multiple spacecraft magnetic measurements utilizes the absolute of (div B/curl B) to evaluate the error. We note that our previous method (NMG, Shen et al., 2021a) for calculating the linear and quadratic magnetic gradients based on 4-spacecraft MMS mission observations has to apply both the div B = 0 and grad div B=0 constraints. Several other methods also utilize the div B = 0 or both the div B = 0 grad div B=0 constraints (Liu et al., 2019; Torbert et al., 2020).

Ref:

Dunlop, M. W., Balogh, A., Glassmeier, K.-H., and Robert, P.: Four-point cluster application of magnetic field analysis tools: The curlometer, J. Geophys. Res., 107, 1384. https://doi.org/10.1029/2001JA005088, 2002b.
Shen, C., Zhang, C., Rong, Z., Pu, Z., Dunlop, M. W., Escoubet, C. P., Russell, C. T.,

Zeng, G., Ren, N., Burch, J. L., Zhou, Y.: Nonlinear magnetic gradients and complete magnetic geometry from multispacecraft measurements, J. Geophys. Res., 126, JA028846. https://doi.org/10.1029/2020JA028846, 2021a.

For the reader it is confusing that the time derivative is used (line 109) in the explanation of the technique, while time derivatives or mixed space-time derivatives do not appear in the variable count on lines 122ff.

Reply:

Here we use the time derivatives and mixed space-time derivatives to add more constraints in order to obtain a unique solution of quadratic gradients from 7- or 9-point observations. We agree that this can lead to confusion as our work is focused on estimation of spatial gradients only. To avoid the confusion, we now specified everywhere whether it is a 'spatial' or 'temporal' gradient where applicable. Also, in our previous version of the manuscript, our assumption involving the temporal change of the magnetic structure was not clearly stated. To better clarify this point, we now emphasized that we assume that the magnetic structures (e.g., flux ropes, current sheets, boundary layers, magnetic reconnection regions, etc.) are slowly evolving during their passages through the multi-point constellations. This assumption has now explicitly been stated in Paragraph 1 of Section 2.

For actual magnetic observations, the time resolution is very high (~0.01sec). There are many data points measured in a short amount of time and therefore it is not difficult to get the time derivative of magnetic field even in high orders. We are concentrated on the spatial gradients of magnetic field in this study. To better clarify these in our manuscript, we added more detailed explanation in Section 2.1.1.

I think having the paragraph from line 122ff in the introductory section would help in setting the broader problem of balancing the number of unknowns versus the number of available observations.

Reply:

The introduction on the system of equations was introduced briefly in the 2nd paragraph of our original manuscript. To better introduce the setting of our problem, we now added some detailed explanation in Paragraph 3 in the introduction section.

The discussion of the volume tensor states that its determinant should be nonzero. At this point, no mention is made of the condition number, which is – practically speaking

– more important than the tensor being non-singular. The statement that "This algorithm requires that the constellation be composed of at least seven spacecraft and that its configuration is non-planar. Because both the 9S/C HelioSwarm and 7S/C Plasma Observatory satisfy these requirements, the linear and quadratic magnetic gradients can be readily obtained" is therefore perhaps a bit optimistic. It is appreciated that in the examples the eigenvalues of the volume tensor are given. Still, that only partially describes the conditioning of the problem.

Reply:

Thanks for the comments. From the equation (12) in the subsection 2.2.1, we have speculated that "The constellation must be nonplanar to achieve this result" without a verification. This could be made clear based on the results obtained in the previous work on high order gradients of physical fields this year (Zhou & Shen, 2024). To clarify our argument, we have now provided a proof as follows.

Following Zhou & Shen (2024), in order the solution exists, it is expected that the position of all the spacecraft in the contellation must not obey the following formula

$$a_{11}\left(x_{(\alpha)}^1\right)^2 + a_{12}x_{(\alpha)}^1 x_{(\alpha)}^2 + a_{12}x_{(\alpha)}^2 x_{(\alpha)}^1 + a_{22}\left(x_{(\alpha)}^2\right)^2 = 0, \tag{1}$$

where $a_{rs}\left(r,s=1,2\right)$ are fixed coefficients. The above equations can be rewritten as

$$a_{11}\left(x_{(\alpha)}^1 / x_{(\alpha)}^2\right)^2 + 2a_{12}\left(x_{(\alpha)}^1 / x_{(\alpha)}^2\right) + a_{22} = 0, \tag{2}$$

which reduce to $x_{(\alpha)}^1 / x_{(\alpha)}^2 = constant$. It means that all the spacecraft are in the plane parallel to the $x_3$ axis or the motion direction. Therefore, it is necessary that the constellation should not be planar in order to deduce the quadratic magnetic gradients as well as the linear magnetic gradient. The next iterations would also require this condition. We have additionally added the following citation.

Ref: Zhou, Y. and Shen, C.: Estimating gradients of physical fields in space, Ann. Geophys., 42, 17–28. https://doi.org/10.5194/angeo-42-17-2024, 2024.

These statements have now been added to the end of Section 2.2.1.

Regarding the examples the eigenvalues of the volume tensor, the three eigenvalues of the volumetric tensor for the random shape of the 7S/C in the tests (see Section 3), are $w_1 = 0.1643 \times 10^{-3} R_E^2$, $w_2 = 0.1104 \times 10^{-3} R_E^2$, and $w_3 = 0.0341 \times 10^{-3} R_E^2$. These are now given at the end of Paragraph 3 of Section 3.

Figure 1 presents a very specific shape of the 7 S/C constellation. Such a constellation is nice for conceptually presenting the idea of "nested tetrahedra", but cannot be easily maintained in space in practice. This figure is nowhere referenced nor discussed.

Reply:

Figure 1 presents a schematic diagram of the Plasma Observatory constellation similar to the mission term proposal. It is indeed not the actual shape of the constellation. Nevertheless, this does not change the generality and applicability of our method. In the tests in section 3, the positions of the seven spacecraft are generated randomly and is illustrated in Figure 2. We note that the shape of the constellation in the tests is not the same as shown in Figure 1.

We have now referred Figure 1 in the paragraph before Eq. (3) with the remark on the constellation shape as mentioned above. It is also noted in the caption of Figure 1.

The effect of measurement errors is not included in the calculation. This is assuming a homogeneous set of instruments, but that may not be the case for Plasma Observatory, for instance, where there are different instruments on mother and daughter spacecraft.

Reply:

We agree with your statement. In this study, we assume that the multi-point magnetic observation can be obtained from identical magnetic measurements from 7 or 9 points simultaneously. The assumption that either space mission having a homogeneous set of instruments has been adopted in both the method and tests. In reality, this could differ especially for Plasma Observatory which remains in the Phase A where the instrument payload is still being optimized. To acknowledge these, we have now added the following statement in Paragraph 2 of the conclusion section.

In this study, simultaneous magnetic measurements from 7 or 9 points were assumed to be able to obtained from identical instruments onboard mothercraft and daughtercraft of the space mission. We also note that the magnetic field data from different detectors need to be synchronized by the time interpolation. In reality, a homogeneous set of instruments onboard spacecraft may not be achieved, and the temporal measurements at different detectors may not be perfectly synchronized.

Regarding the measurement error, the influence of the magnetic detector on the output of method is beyond the scope of our study because it is related to the instrumentation aspect rather than the analytical computation of the magnetic spatial gradient. For the moment, the information on the instruments of HelioSwarm and Plasma Observatory

constellations and their measurement errors are not publicly available. Therefore, it is reasonable to make a restrict evaluation on the measurement errors in the future and after the operations of the missions. We have thus added that "In addition, other systematic errors including measurement errors of the magnetic field are not considered and they are beyond the scope of this study" in the conclusion section.

In addition, we note that in the previous manuscript, we only checked the errors originated from the method itself or the truncation errors.

Regarding the error caused by the measurements, we may make an initial estimation. Starting from the formulas (1) and (2) in the Section 2, we have

$$\Delta f_{(\alpha)} = f_{(\alpha)} - f_c = x^i_{(\alpha)} g_i + \frac{1}{2} x^i_{(\alpha)} x^j_{(\alpha)} G_{ij} \tag{3}$$

Considering the error of the measurement on the positions of the satellite are generally very small and can be neglected, the above equations are linear. It can be expected that the relative errors of gradients are roughly estimated as

$$\frac{\delta g_i}{|g_i|} \sim \frac{\delta f}{|\Delta f|}, \quad \frac{\delta G_{ij}}{|G_{ij}|} \sim \frac{\delta f}{|\Delta f|} \tag{4}$$

Assuming the typical magnetic strength of structure is B, the characteristic spatial scale of the structure is D, while the separation of the satellite is L, then

$$|\Delta f| \sim \frac{LB}{D} \tag{5}$$

So that the relative error is about

$$\frac{\delta f}{|\Delta f|} \sim \frac{\delta B}{LB/D} = \frac{\delta B}{B} \frac{D}{L}, \tag{6}$$

e.g., for the magnetotail measurements, B~20nT, D~2000km, L~200km, $\delta B$~0.01nT. Then the relative error is

$$\frac{\delta B}{B} \frac{D}{L} \approx \frac{0.01nT}{20nT} \cdot \frac{2000km}{200km} \sim 0.005 \tag{7}$$

Therefore, the method may be valid.

Nothing is said about error estimates on the results (in the case where you do not know the exact solution). Does the technique allow you to produce such error estimates? If so, it would be useful to compare these estimates to the actual errors for the two test cases.

Reply:

In addition to our analysis above, we have now expanded Section 4 to better elaborate the error estimates of the various error sources. The divergence and gradient of divergence obtained from algorithm are used to evaluate the error. To offer a uniform standard for evaluation, the divergence and gradient of divergence were non-dimensionalized with the corresponding characteristic quantity.

**Minor comments**

-      Title: Personally, I would try to avoid the "SC" abbreviation in the title. Better change into: "Quadratic Magnetic Gradients from 7- and 9-Spacecraft Constellations"
Reply:

The abbreviation has been deleted in the title and been substituted by full writing accordingly.

line 12: remove "therefore"

Reply:

The correction has been made accordingly. Thanks.

line 13: from -> from the

Reply:

The correction has been made accordingly. Thanks.

line 17: The tests -> Tests

Reply:

The correction has been made accordingly. Thanks.

line 18: verifies -> verified

Reply:

The correction has been made accordingly. Thanks.

line 23: iteration algorithm -> iterative algorithm

Reply:

The correction has been made accordingly. Thanks.

line 38: gradient -> gradients

Reply:

The correction has been made accordingly. Thanks.

line 43: tetrahedral -> a tetrahedral

Reply:

The correction has been made accordingly. Thanks.

line 43: such the missions -> such missions

Reply:

The correction has been made accordingly. Thanks.

line 62: consisting -> consisting of

Reply:

The correction has been made accordingly. Thanks.

line 66: an ESA's new mission -> a new ESA mission

Reply:

The correction has been made accordingly.

line 68: drawn -> inferred

Reply:

The correction has been made accordingly. Thanks.

line 74ff: I suggest to change punctuation into: "a description of the tests conducted for two typical magnetic structures (a cylindrical force-free flux rope and a dipole magnetic field), which were utilized to check the validity and accuracy of the new algorithm, is given …"

Reply:

Punctuations have been changed accordingly. Thanks.

line 76: error -> accuracy

Reply:

The correction has been made accordingly. Thanks.

line 84: references -> reference frames

Reply:

The correction has been made accordingly. Thanks.

line 84: … of the magnetic field

Reply:

The correction has been made accordingly.

caption of Figure 1: relative to the constellations -> relative to the constellation

Reply:

The correction has been made accordingly.

line 95, 97, 195 and elsewhere: no capital needed at the beginning of the line

Reply:

Corrections have been made accordingly.

line 99: draw -> infer

Reply:

The correction has been made accordingly. Thanks.

line 225: "The characteristic size of the S/C is twice the square root of the maximum eigenvalue" makes no sense. Size of the S/C constellation?
Reply:

Here we follow the definition of Harvey (1998) on the "characteristic size" in the tool book (Section 12.4.3). The characteristic size is used to indicate the size of the polyhedron and is twice the square root of the maximum eigenvalue of the volumetric tensor.

Ref: Harvey, C. C.: Spatial gradients and the volumetric tensor, in: Analysis methods for multi-spacecraft data, edited by Paschmann, G. and Daly, P. W., European Space Agency Publ. Division, Noordwijk, Netherlands, 315-319, 1998.

Fig 4, 6, 8: light yellow lines are hardly visible
Reply: Color of lines in the figures have been changed accordingly. Thanks.

explain abbreviations when first used: NASA, ESA

Reply:

Abbreviations have been explained accordingly. Thanks.

Replies to the second reviewer

Dear Editor,

We thank the reviewer for the valuable comments and suggestions, which benefit significantly the improvements of the paper. The main change includes the discussion of the various sources of errors as pointed out by the referee. We address the referee's point-by-point below and improve our manuscript presentation throughout.

Since we solicited a significant amount of expertise from a colleague to improve this manuscript, we have added him as a co-author to this revised version of the manuscript. In this reply, the comments of the referee are marked in black and the replies in blue. We hope that our revised manuscript is now more accessible to the general readers.

Yours sincerely,

Chao Shen, on behalf of the co-authors.
* * *
Comments on the manuscript entitled

Quadratic Magnetic Gradients from 7-SC and 9-SC Constellations

submitted by Chao Shen, Gang Zeng, and Rungployphan Kieokaew.

**General comments**

The manuscript is concerned with a novel method to estimate the first and second spatial derivatives of stationary magnetic structures from multi-point measurements in constellations consisting of N spacecraft where N=7 (Plasma Observatory) or N=9 (HelioSwarm). In addition to the set of 3N simultaneous magnetic field measurements, also discrete representations of the 3N first time derivatives are utilised in the method, amounting to an effective number of 6N input data that are used for estimating the 33 model parameters (30 parameters in the second-order Taylor expansion, and 3 parameters for the velocity of the stationary structure). The paper presents the model equations and an iterative algorithm for estimating the parameters. The method is demonstrated using two magnetic field models. Deviations of the model predictions from their analytical counterparts are discussed.

While the study presented here can be considered a proof of concept that introduces the general framework and demonstrates the processing flow of the proposed method, a number of open issues and limitations need to be addressed and critically discussed, e.g., the concept of stationarity in the context of magnetohydrodynamics, the different types of errors, and the numerical stability of the inversion/reconstruction method.

Stationarity in the context of magnetohydrodynamics:

The method utilises discrete time derivative measurements through advection-type equations (3): $\frac{\partial \mathbf{B}}{\partial t} = -\mathbf{V} \cdot \nabla \mathbf{B}$ . In magnetohydrodynamics, however, the local time derivative of the magnetic field $\mathbf{B}$ is connected to the velocity $\mathbf{V}$ through Faraday's law and an appropriate Ohm's law, which in the ideal case (collision-free plasmas in geospace and the heliosphere) equates the local time derivative with the curl of the cross product of velocity and magnetic field: $\frac{\partial \mathbf{B}}{\partial t} = \nabla \times (\mathbf{V} \times \mathbf{B})$ (hydromagnetic theorem), implying the invariance of magnetic flux through a surface transported with the plasma flow. The authors are asked to explain in which sense their notion of stationarity differs from the canonical interpretation (frozen-in magnetic flux) in space plasma physics.

Reply:

Thank you for raising this interesting point. For the ideal MHD fields, the following magnetic convection equation is valid:

$$\frac{\partial \vec{B}}{\partial t} = \nabla \times (\vec{u} \times \vec{B}),$$ (1)

   or

$$\frac{\partial \vec{B}}{\partial t} = \vec{u} \cdot \nabla \vec{B} + \vec{B} \cdot \nabla \vec{u} - \vec{B}(\nabla \cdot \vec{u}),$$ (2)

where $\vec{u}$ is the bulk velocity of the MHD plasmas.

As presented in the manuscript, another relationship is valid:

$$\frac{\partial \vec{B}}{\partial t} = -(\vec{v} \cdot \nabla)\vec{B},$$ (3)

where $\vec{v}$ is the apparent velocity of the magnetic structure.

At the first glance, $\vec{u}$ and $\vec{v}$ are possibly the same. However, for some situations, $\vec{u}$ and $\vec{v}$ are different. For example, when we observe the shock wave front at the shock frame, we see that the shock is at rest with zero apparent velocity, while the upstream and downstream plasmas are moving at their bulk velocities.

For some cases, it is possible that $\vec{u} = \vec{v}$, e.g., in magnetic clouds. We may check this kind of situation when $\vec{u} = \vec{v}$. Combining the Eqs. (2) and (3) yields

$$(\vec{B}\cdot\nabla)\vec{v} - \vec{B}\nabla\cdot\vec{v} = 0, \tag{4}$$

which limits the fluid velocity.

Furthermore, if the MHD plasma is incompressible, then

$$\nabla\cdot\vec{v} = 0 \tag{5}$$

so that,

$$(\vec{B}\cdot\nabla)\vec{v} = 0 \tag{6}$$

which means the velocity of the plasma is constant along the magnetic field lines.

The only situation we could expect to satisfy both Eqs. (5) and (6) is

$$\vec{v} = \text{constant} \tag{7}$$

everywhere. This implies that the MHD fluids are in an equilibrium state. Therefore, it is most likely that $\vec{u}$ cannot be equal to $\vec{v}$ in all situations except that the MHD plasmas are at equilibrium.

In brief, we emphasize that the Eq. (3) is valid for various space plasmas unlimited to MHD plasma. As we addressed a comment of the first referee, we have now noted in Section 2.1 that "In these limits, we assume that the magnetic structures are slowly evolving during their passages through the multi-point constellations such that any differences in the measurements at different spacecraft can be attributed to the spatial variations rather than the temporal changes (i.e., evolution of magnetic structures)."

The constraints to the Eq. (3) are that the plasmas are highly conductive and have a very low velocity ($v/c \ll 1$, where c is the speed of light in vacuum), and the physical processes are slowly evolving at low frequencies. This statement has now been added in the paragraph below Eq. (3) in Section 2.1.

Discretisation errors:

As pointed out by the authors in lines 223-227, the separation of spacecraft in the array introduces up to three different spatial discretisation scales. It should be added that a fourth spatial scale comes into play through the finite difference representation of local time derivatives, namely, the product of the intrinsic time scale (time resolution) with the velocity of the magnetic structure in the spacecraft frame.

Reply:

Thanks for pointing this out. In general, the separation of the spacecraft in a constellation is several 100km to several 1000km. For the time dimension, the time resolution of the magnetic measurement $\Delta t$ is about 0.01 sec, i.e., $\Delta t = 0.01\text{sec}$. Considering that the magnetic structure is moving at a velocity V<500 km/s, the spatial

resolution along the motion direction is about $v\Delta t < 5km$, which is much less than the S/C separation. Therefore, it can be excepted that the error brought will be much less. In brief, we expect that the discretization error would be minimal. The above analysis has now been added as a new paragraph of Section 4.

Iteration errors:

When in Section 3 the convergence properties of the iterative method are discussed, a particular type of error considered there is the mismatch of the actual limit of the procedure and the approximation reached after a finite number of iterations. This error may be termed iteration error. It is not associated with the finite resolution of the spacecraft array or the time series and thus needs to be considered separately.

Reply:

In this work, we employ an iterative procedure to solve the problem. The transformation relationship (3) is used as the constraints. It is noted that both the two Eqs. in the formula (3) in the text are nonlinear with a 2nd-order term on the left-hand side. However, the iterative procedures have made the problem a linear one. This helps reduce the calculation error and make the calculation more stable. Nevertheless, it is not easy to get the formula of the error of the iterations. As shown in Sections 3.1 and 3.2, we performed two tests on the typical magnetic fields to illustrate the feasibility of this method and check the errors. The detailed evaluation on the iteration accuracy of this algorithm can be made in the future when the real mission data are available. This discussion has now been added as a new paragraph 3 in Section 4.

Random errors:

Due to imperfect (noisy) input data (measurement inaccuracies), the estimated parameters (first and second derivatives) will be subject to random errors, in addition to the discretisation errors and iteration errors mentioned above. In the current version of the manuscript, with demonstrations using noise-free model magnetic fields only, neither random errors are considered, nor the stability of the estimation (inversion) procedure (parameter reconstruction from noisy input data) which is likely to be associated with the set of different spatial discretisation scales. Since the inverse problem is weakly nonlinear, a condition number could be constructed for the linearised problem in the iterative procedure, or Monte Carlo simulations could be utilised to assess the impact of random errors. If such an approach is considered beyond the scope of this paper, the authors should at least critically discuss the implications of random errors, and outline the directions for future work.

Reply:

The noise or disturbances in the data can come from the measurement error or the presence of high-frequency (physical) fluctuations such as those from plasma waves. This can make the calculation of the high order magnetic gradients very difficult (see Shen et al. 2021). When analyzing the actual observation data, filtering methods should be employed to remove the high frequency components and avoid the negative effect of the noise. This process would help to extract large-scale magnetic structures under the consideration. We acknowledge that the measurement errors are another source of errors that can influence our spatial linear and quadratic gradient estimation. This discussion has been added as a new paragraph 4 of Section 4.

Magnetic field divergence:

To quantitatively assess the limitations of this high-dimensional reconstruction problem with 33 model parameters, it is not sufficient to consider only a scalar quantity such as an estimate of the divergence of the magnetic field. Furthermore, in its original form, the divergence is normalised by the curl of the magnetic field (lines 201-203), while the latter quantity is zero for one of the two test cases (dipole field) in Section 3. To see if a dimensionless version of the divergence differs significantly from zero, meaningful reference values need to be chosen.

Reply:

1. In this approach, the magnetic Gauss's law ($\nabla \cdot \vec{B} = 0$ along with $\nabla\left(\nabla \cdot \vec{B}\right) = 0$)

   has been used as the measures of the errors of the first order and second order magnetic gradients for the actual data analysis. Really it is not perfect because it can not include partial components of the magnetic gradients ( the formula $\nabla \cdot \vec{B} = 0$

   contains 3 of the total 9 components of $\nabla\vec{B}$ while $\left|\nabla\left(\nabla \cdot \vec{B}\right)\right| = 0$ contains 9 of

   the total 18 components of $\nabla\nabla\vec{B}$). The advantage to use them as the measures of the errors of the magnetic gradients is that they are robust and also simple. We still have not found other better ways for evaluating the accuracy of the algorithm because the actual values of the magnetic gradients are unknown for comparation when analyzing the real observation data. This discussion has been added to lines 425 – 427 in Section 4.
2. It is true that the method for calculating the error in the Curlometer method is invalid when there is no electric current. Here we use the normalized forms to avoid this problem.
3. It is certain that the characteristic magnetic field and spatial scale of the structures must be properly chosen during the actual data analyses thus the errors resulted can

represent the accuracy of the calculations as evidenced in Figures 12 and 13 and discussed in the text.

Terminology:

It is very unusual to refer to the tensor of second partial derivatives as the "quadratic gradient". It is strongly recommended to adjust the terminology. Canonical options are: "Hessian" or "Hessian matrix" (2nd derivatives of a scalar field) or "Hessian tensor".

Reply:

We are also very concerned of the names of the second partial derivatives. Hessian tensor is a possible name for it, but too unfamiliar to the average readers. Liu et al. (2019) have used the second order gradient for it, which are somewhat too long if frequently used. Torbert et al. (2020) have used quadratic coefficient for it. In the 1998 data analysis book, Chanteur (the first to stress this question) has used the term quadratic for it. Therefore, we think that the cautious and proper way may be calling it quadratic gradient, and giving a note of Hessian tensor for it at the beginning (in the Introduction section). It is noted that the second order magnetic gradient is composed of 18 components, while Hessian tensor contains 6 components because it commonly means the second partial derivatives of a scalar. The term "Hessian matrix" has been added in a bracket in the 3rd paragraph in the introduction section.

**Specific comments**

Abstract and Key Points:

- The statements
"The tests for the situations of magnetic flux ropes and dipole magnetic field have verifies the validity and accuracy of this approach."
and
"Magnetic flux ropes and dipole magnetic field testing verifies the validity and accuracy of the approach."
are too strong (and also difficult to understand in the first place). A proof of concept is presented, but a complete assessment of the accuracy would require studying all error types and the stability of the model inversion procedure.

Reply:

As the referee pointed out, not all error types have been considered. This manuscript only evaluates the truncation errors. It is the limitation of our study. The strong statements have been modified accordingly.

The first reviewer also raised the problem on the measurement errors, please refer to the reply to the referee #1.

Theoretical evaluating on iteration stability is a tough work, which cannot be solved completely in a short time. In this initial study, we are concentrated on the feasibility of the algorithm. Nevertheless, two tests made in this research have confirmed the reliability of the method because the iterations for both tests can arrive at convergence and the total errors are rather small.

Introduction:

- Line 54: The statement "To obtain high-order gradients in the magnetic field ..." is ambiguous as it could also refer to different orders of accuracy in discrete representations of the gradient. Instead, one could write "To estimate second derivatives of the magnetic field ..."

Reply:

The correction has been made accordingly. Thanks.

Method:
- Line 81: The statement "Calculation of the linear and quadratic gradients of a magnetic field generally requires magnetic measurements from at least ten spacecraft" should be made precise and briefly explained (3+9+18=30 parameters in the Taylor expansion up to second order, 3N magnetic field measurements in an array with N spacecraft).

Reply:

The problem of balancing the number of unknowns versus the number of available observations has now been elaborated in the first paragraph of Section 2.1. A more elaborated explanation has been also added in paragraph 3 of the introduction section.

- Lines 105/106: The statement "The errors in formula (3) are on the order V/c." is unclear. What kind of errors? Meaning of the variable c? Non-relativistic limit?

Reply:

The errors mean the truncation errors of the formula (3) compared with the accurate one. Here V is the apparent speed of the magnetic structure and c is the speed of light in vacuum, which are explained in the text now. In the non-relativistic limit (V/c<<1, it is generally valid in space plasmas) we can derive the simple formula (3). This explanation has now been added in the paragraph below Eq. (3). Indeed, the accurate formula is complicated for the analysis. An explanation on this issue is in the talk of the first author in the EGU meeting this year and the ppt is attached for the reference.

- Line 114: first-order or zero-order?

Reply:

In the referee mentioned line, it is first-order. But in the second line below, it should be also first-order. The correction has been made accordingly. Thank you.

- Line 121: In the statement "The iterations are performed repeatedly until satisfactory results are achieved.", quantify what is meant by "satisfactory results" (which error measure/threshold).

Reply:

As shown in Figures 4 and 8, the errors become smaller and smaller with increasing number of iterations, which means that the iterative results are convergent. In the tests, the number of iterations is set to 100, and the results with 100 iterations are regarded as "satisfactory results". The sentence has been modified as "The iterations are performed repeatedly until results are converge, which means satisfactory results are achieved."

- Line 142: In the statement "The temporal variation rate ... is readily obtained using time-series magnetic observation.", explain how the temporal variation rate is approximated (finite differencing? time resolution?).

Reply:

In the test, central difference has been used. And the first value of series data has been obtained by first order forward difference, and the last value has been obtained by first order backward difference. Explanation has been made after the sentence accordingly.

For actual magnetic observations, the time resolution is very high (~0.01sec), there are plenty of time series data, so it is not difficult to get the time derivative of magnetic field even in high orders.

Comparison of new method with analytical modelling:

- General comment: With spacecraft separations on the order 0.01 RE, and model magnetic fields varying on spatial scales on the order RE, the magnetic configurations vary only gradually on the spacecraft array scale, so these are not particularly challenging tests of the proposed method. In geospace, magnetic field structures can vary on much smaller scales. Furthermore, the model magnetic field configurations are simplified and highly symmetrical structures with a very small number of parameters so that only a minor subset of the 33 degrees of freedom can be assessed. The specifics and the limitations of the chosen test cases should thus be critically discussed.

Reply:

The linear gradient of the magnetic field has 9 components, while the quadratic gradients comprise 18 independent components due to the symmetry of quadratic gradients. For the flux rope case, only 3 components of linear gradient and 5 components of quadratic gradients have been assessed. But for dipole field case, 4 components of linear gradient and 10 components of quadratic gradients have been assessed. The number of assessed parameters has reached half. We have chosen so symmetrical model magnetic field in order to easily compare the simulation results with the accurate analytic calculations. Nevertheless, these are still somewhat complete tests because the zero components of the magnetic gradients are calculated with the algorithm as well and checked. Accordingly, further evaluations on the algorithm could be made with the modeled magnetosphere with less symmetry in the future. This discussion has now been added as a new Section 3.3.

- Lines 226/227: With the given value of the first eigenvalue $w_1 = 0.1643\, R_E^2$, the characteristic size L should be $L = 2\sqrt{w_1} = 0.8106\, R_E$.

Reply:

In the manuscript, the value of characteristic size L is correct, but wrong values of the eigenvalues are given. $w_1 = 0.1643 \times 10^{-3} R_E^2$, $w_2 = 0.1104 \times 10^{-3} R_E^2$, and

$w_3 = 0.0341 \times 10^{-3} R_E^2$. Corrections have been made accordingly. Thanks.

- Lines 254-258: Only total errors after a given number of iterations are discussed. It would be more interesting to get separate assessments of iteration errors and discretisation errors.

Reply:

It is not very easy to separate iteration errors and discretisation errors. The separate assessments of errors should be considered in future work.

- Lines 277/278 and line 321: In the statement "The relative error approaches 50%; however, the absolute error is low." it is not clear which reference is used (low/small compared to what?)

Reply:

The term "low" is compared to zero. Explanation has been made accordingly. "however, the absolute error is just 0.143, which is approaching zero."

Errors:

- General comment: As explained above, this section is very incomplete regarding the various types of errors. In particular, the current version of the manuscript lacks a critical discussion of random errors and the stability of the parameter estimation (inversion) procedure.

Reply:

Regarding the stability of the parameter estimation (inversion) procedure, refer to the above response.

Regarding the random errors: The noise or disturbances in the data can come from the measurement error, but they could mainly be caused by the plasma waves. This can make the calculation of the high order magnetic gradients very difficult (see Shen et al. 2021). In analyzing the actual observation data, we could use filtering methods to

remove the high frequency components so as to smooth the raw data and avoid the negative effect of the data disturbance at utmost.

- Figures 12 and 13: It may be worth mentioning that the errors of the first derivative decrease quadratically with the scale L (second-order accuracy with regard to discretisation errors) whereas the errors of the second derivatives decrease linearly with L (first-order accuracy with regard to discretisation errors).

Reply:

[Figure]

The above two figures show the trend of dimensionless divergence and gradient of divergence with characteristic size L for dipole field and flux rope case, respectively. As referee's suggestion, the errors of the first derivative decrease quadratically with the scale L whereas the errors of the second derivatives decrease linearly with L. This conclusion has been added in the manuscript accordingly.

Conclusions:

- General comment: In line with the previous comments, this section should be rewritten to reflect the actual limitations of this study and the method, and explain where further work is required.

Reply:

Thanks for pointing this out. In the future, the random errors, measurement errors, iteration errors and discretisation errors could be evaluated in details, especially when the mission payloads are fixed and the real mission data are available. The statement has been added in Conclusions (Section 5) accordingly.

**Appendix**

**Transformations of electromagnetic fields in different reference frames and their applications**

**Chao Shen**

Harbin Institute of Technology, Shenzhen, China

**Yong Ji**

Ningxia University, Yinchuan, China

2024.10.15,EGU,Vienna

**1.   Background**

The transformation of electromagnetic fields across different frames of reference, whether inertial or non-inertial, is a common challenge encountered in electromagnetic space measurements and analyses.

[Figure]

[Figure]

[Figure]

[Figure]

**2. Lorentz transformations in low-speed moving reference frames**

**2.1 space-time transformation**

The impact of the equivalent gravity on the non-inertial reference system is $2\varphi / c^2 \sim u^2 / c^2 \ll 1$ ,

the physical laws for flat spacetime remain valid.

[Figure]

Space-time coordinates in frame K: $(x^\mu) = \left(x^0 = ct, \mathbf{x}\right)$

Space-time coordinates in frame K': $(x'^\mu) = \left(x'^0 = ct', \mathbf{x}'\right)$

Lorentz transformations for space-time coordinates

$$dx'^\mu = \frac{\partial x'^\mu}{\partial x^\nu} dx^\gamma = \Lambda^\mu_\gamma dx^\gamma \qquad \frac{\partial}{\partial x'^\mu} = \frac{\partial x^\nu}{\partial x'^\mu} \frac{\partial}{\partial x^\nu} = \hat{\Lambda}^\nu_\mu \frac{\partial}{\partial x^\nu}$$

Lorentz transformation matrix

$$\Lambda^\mu_\gamma = \frac{\partial x'^\mu}{\partial x^\gamma} = \begin{pmatrix} \gamma & -\gamma\beta_j \\ -\gamma\beta_i & \delta_{ij} + \frac{\gamma^2}{\gamma+1}\beta_i\beta_j \end{pmatrix} \qquad \hat{\Lambda}^\mu_\gamma = \frac{\partial x^\mu}{\partial x'^\gamma} = \begin{pmatrix} \gamma & \gamma\beta_j \\ \gamma\beta_i & \delta_{ij} + \frac{\gamma^2}{\gamma+1}\beta_i\beta_j \end{pmatrix} \qquad \begin{array}{l} \beta_i = u_i / c \\ \gamma = \left(1 - \beta^2\right)^{-\frac{1}{2}} \end{array}$$

**Local transformation of the spacetime coordinate**

$$d\mathbf{x}' = d\mathbf{x} + \frac{\gamma^2}{(\gamma+1)c^2}(d\mathbf{x}\cdot\mathbf{u})\mathbf{u} - \gamma\mathbf{u}dt$$

$$dt' = \gamma\left(dt - \mathbf{u}\cdot d\mathbf{x}/c^2\right)$$

Superposition law of velocities

$$\mathbf{v}' = \frac{1}{1-\mathbf{v}\cdot\mathbf{u}/c^2}\left[\frac{1}{\gamma}\mathbf{v} - \mathbf{u} + \frac{\gamma}{\gamma+1}\cdot\frac{(\mathbf{v}\cdot\mathbf{u})\mathbf{u}}{c^2}\right]$$

4-vector potential $(\phi/c, \mathbf{A})$, 4-current density, $(c\rho, \mathbf{j})$

4-wave vector $(\nu/c, \mathbf{k})$.......

Local Lorentz transformation

$$p'^{\mu} = \Lambda^{\mu}_{\gamma}p^{\gamma}$$

The transformation of the electromagnetic tensor

$$F'^{\mu\nu} = \Lambda^{\mu}_{\lambda}\Lambda^{\nu}_{\sigma}F^{\lambda\sigma}$$

**The transformations of the electro-magnetic fields for non-inertial reference frames**

$$\left\{ \begin{array}{l} E' = \gamma\left( E + u\times B - \dfrac{\gamma}{\gamma+1}(u\cdot E)u/c^2 \right) \\[4mm] B' = \gamma\left( B - \dfrac{1}{c^2}u\times E - \dfrac{\gamma}{\gamma+1}(u\cdot B)u/c^2 \right) \end{array} \right.$$

$$\left\{ \begin{array}{l} \phi' = \gamma\left( \phi - u\cdot A \right) \\[4mm] A' = A - \dfrac{\gamma^2}{c^2}\phi u + \dfrac{\gamma^2}{(\gamma+1)c^2}(u\cdot A)u \end{array} \right.$$

$$\left\{ \begin{array}{l} P' = \gamma\left( P - \dfrac{1}{c^2}u\times M - \dfrac{\gamma}{\gamma+1}(u\cdot P)u/c^2 \right) \\[4mm] M' = \gamma\left( M + u\times P - \dfrac{\gamma}{\gamma+1}(u\cdot M)u/c^2 \right) \end{array} \right.$$

$$\left\{ \begin{array}{l} \rho' = \gamma\left( \rho - u\cdot j/c^2 \right) \\[4mm] j' = j - \gamma\rho u + \dfrac{\gamma^2}{(\gamma+1)c^2}(u\cdot j)u \end{array} \right.$$

$$\left\{ \begin{array}{l} v' = \gamma\left( v - u\cdot k \right) \\[4mm] k' = k - \dfrac{\gamma}{c}ku + \dfrac{\gamma^2}{(\gamma+1)c^2}(u\cdot k)u \end{array} \right.$$

The errors are of the second order of u/c.

Lorentz transformation matrix under the first-order approximation

$$(\Lambda^{\mu}_{\nu}) = \begin{pmatrix} 1 & -\beta_j \\ -\beta_i & \delta_{ij} \end{pmatrix}$$

The transformation of space-time coordinates for low speed cases

$$\begin{cases} d\mathbf{x}' = d\mathbf{x} - \mathbf{u}dt \\ dt' = dt - \dfrac{1}{c^2}\mathbf{u} \cdot d\mathbf{x} \end{cases}$$

The transformations of the electro-magnetic fields

$$\begin{cases} \phi' = \phi - \mathbf{u} \cdot \mathbf{A} \\ \mathbf{A}' = \mathbf{A} - \frac{1}{c^2}\phi\mathbf{u} \end{cases} \qquad \begin{cases} \mathbf{E}' = \mathbf{E} + \mathbf{u} \times \mathbf{B} \\ \mathbf{B}' = \mathbf{B} - \frac{1}{c^2}\mathbf{u} \times \mathbf{E} \end{cases} \qquad \begin{cases} \boldsymbol{P}' = \boldsymbol{P} - \dfrac{1}{c^2}\boldsymbol{u} \times \boldsymbol{M} \\ \boldsymbol{M}' = \boldsymbol{M} + \boldsymbol{u} \times \boldsymbol{P} \end{cases}$$

$$\begin{cases} \rho' = \rho - \mathbf{u} \cdot \mathbf{j}/c^2 \\ \mathbf{j}' = \mathbf{j} - \rho\mathbf{u} \end{cases} \qquad \begin{cases} \nu' = \nu - \mathbf{k} \cdot \mathbf{u} \\ \mathbf{k}' = \mathbf{k} - \frac{1}{c}k\mathbf{u} \end{cases}$$

The errors are of the second order of u/c.

**3. Transformations for electromagnetic fields in space plasmas**

**3.1 Galilean transformation for spacetime**

Local Galilean transformation

$$\begin{cases} d\mathbf{x}' = d\mathbf{x} - \mathbf{u}dt \\ dt' = dt \end{cases}$$

Spacetime transform matrix

$$(\Lambda^{\mu}_{\nu}) = \begin{pmatrix} 1 & 0 \\ -\beta_j & \delta_{ij} \end{pmatrix}$$

Galilean velocity superposition    $\mathbf{v}' = \mathbf{v} - \mathbf{u}$

The transformations of spacetime gradients

$$\begin{cases} \dfrac{\partial}{\partial t'} = \dfrac{\partial}{\partial t} + \mathbf{u} \cdot \nabla_{\mathbf{x}} \\ \nabla_{\mathbf{x}'} = \nabla_{\mathbf{x}} \end{cases} \qquad\qquad \begin{cases} \dfrac{\partial}{\partial t} = \dfrac{\partial}{\partial t'} - \mathbf{u} \cdot \nabla_{\mathbf{x}'} \\ \nabla_{\mathbf{x}} = \nabla_{\mathbf{x}'} \end{cases}$$

Total derivative    $\dfrac{D}{dt} \equiv \dfrac{\partial}{\partial t} + \mathbf{v} \cdot \nabla_{\mathbf{x}}$    satisfies    $\dfrac{D'}{dt'} = \dfrac{D}{dt}$

**3.2 Galilean transformations for an electromagnetic field**

Due to the high conductivity of space plasmas,
The transform matrix for electromagnetic field is

$$(\Lambda^{\mu}_{\ \nu}) = \begin{pmatrix} 1 & -\beta_j \\ 0 & \delta_{ij} \end{pmatrix}$$

The transformations of the electro-magnetic fields

$$\begin{cases} \phi' = \phi - \mathbf{u} \cdot \mathbf{A} \\ \mathbf{A}' = \mathbf{A} \end{cases} \qquad \begin{cases} \mathbf{E}' = \mathbf{E} + \mathbf{u} \times \mathbf{B} \\ \mathbf{B}' = \mathbf{B} \end{cases} \qquad \begin{cases} \mathbf{P}' = \mathbf{P} - \dfrac{1}{c^2}\mathbf{u} \times \mathbf{M} \\ \mathbf{M}' = \mathbf{M} \end{cases}$$

$$\begin{cases} \rho' = \rho - \mathbf{u} \cdot \mathbf{j} / c^2 \\ \mathbf{j}' = \mathbf{j} \end{cases} \qquad \begin{cases} \nu' = \nu - \mathbf{u} \cdot \mathbf{k} \\ \mathbf{k}' = \mathbf{k} \end{cases}$$

For each group of Eqs., the error for the 1st Eq is of the second order of u/c, and that for the 2nd Eq. is of the first order of u/c.

**3.3 Galilean transformations for the spatial and temporal gradients of the electromagnetic field**

The gradients of magnetic field are invariants for different frames

$$\nabla'\mathbf{B}' = \nabla\mathbf{B} \qquad \nabla'\nabla'\mathbf{B}' = \nabla\nabla\mathbf{B} \qquad \nabla'\nabla'\nabla'\mathbf{B}' = \nabla\nabla\nabla\mathbf{B}$$

Relationships between the spatial and temporal gradients of the electromagnetic field

$$\begin{cases} \partial_t \mathbf{B} = -\mathbf{u} \cdot \nabla\mathbf{B} \\ \partial_t \nabla\mathbf{B} = -\mathbf{u} \cdot \nabla\nabla\mathbf{B} \end{cases}$$

These formulas are commonly used to calculate the structures' relative velocities and partial components of the magnetic gradients.

For low-speed space plasmas, the Coulomb gauge is valid:

$$\nabla \cdot \mathbf{A} = \nabla' \cdot \mathbf{A}' = 0$$

**4. Application 1: Corotational potential of planets with intrinsic magnetic fields**

The dipole magnetic field of a planet

$$\mathbf{B}' = \frac{3\mathbf{r}'(\mathbf{r}'\cdot\mathbf{M}) - r'^2\mathbf{M}}{r'^5} \qquad \mathbf{A}' = \frac{\mathbf{M}\times\mathbf{r}'}{r'^3}$$

[Figure]

Corotation electric field observed in the planetary centroid reference frame

$$\mathbf{E}_{cor} = -\mathbf{u}\times\mathbf{B}' = -(\mathbf{\Omega}\times\mathbf{R})\times\mathbf{B}'$$

The corotation potential

$$\phi_{cor} = \mathbf{u}\cdot\mathbf{A}' = \frac{1}{r'^3}\left[(\mathbf{M}\cdot\mathbf{\Omega})(\mathbf{R}\cdot\mathbf{r}') - (\mathbf{M}\cdot\mathbf{R})(\mathbf{\Omega}\cdot\mathbf{r}')\right]$$

If the magnetic core coincides with the planetary centroid, and the magnetic moment is aligned with the rotation axis, then

$$\phi_{cor} = \frac{\Omega M \sin^2\theta'}{r'} = \frac{\Omega M}{LR_p}$$

(Note: the divergence is not strictly zero: $\nabla\cdot\mathbf{E}_{cor} = 2\mathbf{B}'\cdot\mathbf{\Omega}$ )

The error is of the first order of u/c

The analysis involves transformations among four different reference frames:

[Figure]

(1) Spacecraft rotational reference frames $\tilde{K}$: $\mathbf{E} = -\nabla_{\tilde{\mathbf{r}}}\phi - \partial_{\tilde{t}}\mathbf{A}$

(2) Spacecraft centroid reference frames $K_s$: $\mathbf{E}_s = -(\nabla_s\phi) - \frac{\partial}{\partial t_s}\mathbf{A_s} = \bar{\mathbf{E}}_s - \frac{\partial}{\partial t_s}\mathbf{A_s}$

(3) Constellation centroid reference frame $K$: $\mathbf{E} = \mathbf{E}_s - \mathbf{v}\times\mathbf{B} = \bar{\mathbf{E}} - \frac{\partial}{\partial t_s}\mathbf{A_s}$

Charge density: $\rho = \varepsilon_0\nabla\cdot\mathbf{E} = \varepsilon_0\nabla\cdot\bar{\mathbf{E}}$

(4) Proper reference frame of the structure $K'$:

$$\rho' = \rho - \frac{1}{c^2}\mathbf{V}\cdot\mathbf{j} = \varepsilon_0\left(\nabla\cdot\bar{\mathbf{E}}\right)_c - \frac{1}{c^2}\mathbf{V}\cdot\mathbf{j}$$

The error is at the first order of u/c.

**6. Conclusions**

* **(1) The Lorentz transformation is suitable for electromagnetic field transformations between any low-speed reference frames, whether inertial or non-inertial.**

* **(2) Approximate transformations for electromagnetic fields in various reference frames for low-speed, slow-varying space plasmas.**

* **(3) A general formula for the planetary corotation electric field is presented.**

* **(4) verifying the deduction of charge density from MMS electrostatic field measurements**

**Thank You for your Attention !**

**3.4 Eqs of MHD obey Galilean transformations**

Equations of MHD contains

$$\frac{D\rho_m}{dt} + \rho_m \nabla \cdot (\boldsymbol{v}) = 0 \qquad\qquad \rho_m \frac{D\boldsymbol{v}}{dt} = \boldsymbol{J} \times \boldsymbol{B} - \nabla p$$

$$\nabla \times \boldsymbol{B} = \mu_0 \boldsymbol{J} \qquad\qquad \nabla \times \boldsymbol{E} = -\frac{\partial \boldsymbol{B}}{\partial t}$$

$$\nabla \cdot \boldsymbol{B} = 0 \qquad\qquad \nabla \cdot \boldsymbol{E} = \varepsilon_0^{-1} \rho_e$$

$$\nabla \cdot \boldsymbol{J} = 0 \qquad\qquad \boldsymbol{J} = \sigma_0 (\boldsymbol{E} + \boldsymbol{v} \times \boldsymbol{B})$$

They are invariant under the following Galilean transformations :

$$\mathbf{v'} = \mathbf{v} - \mathbf{u} \qquad \frac{\partial}{\partial t'} = \frac{\partial}{\partial t} + \mathbf{u} \cdot \nabla_{\mathbf{x}} \qquad \nabla_{\mathbf{x'}} = \nabla_{\mathbf{x}} \qquad \frac{D'}{dt'} = \frac{D}{dt}$$

$$\mathbf{B'} = \mathbf{B} \qquad \mathbf{E'} = \mathbf{E} + \mathbf{u} \times \mathbf{B} \qquad \mathbf{J'} = \mathbf{J} \qquad \rho_e' = \rho_e - \mathbf{u} \cdot \mathbf{J} / c^2$$

---

## Author Response (AR2)

Replies to the first Reviewer

We are very grateful for the referee's valuable comments. We address your suggestions point-by-point below and improve our manuscript presentation throughout. Major changes include the evaluation of discretisation error, iteration error, and measurement error.

In this reply, the comments of the referee are marked in black and the replies in blue. We hope that our revised manuscript is now more accessible to the general readers.

The authors have made a serious effort in responding to the reviewer comments and this has improved the manuscript considerably. There are still a number of language problems (see below). The manuscript might benefit from additional language editing.

Reply:

We thank the referee for these valuable comments. In this version, we have improved the structure of the manuscript throughout, particularly in the evaluation of discretisation error, iteration error, and measurement error. Please see our replies to your major comments point-by-point below.

**Comments**

In section 4 there is a discussion of the errors, which is very much appreciated. The authors talk about truncation error, discretisation error, iteration error, measurement error and random error. Perhaps a brief recollection of the definitions of each of these would be in place here.

Reply:

The definitions of these errors were in fact mentioned in paragraphs 2 – 4 in section 4. To make them clear, we have now spelled out the names of the errors – the discretisation error, iteration error and measurement error or random error – in the first paragraph of section 4. We additionally added in the first paragraph that we briefly introduce them (in the following paragraphs) before analysing them analytically in Appendix A-C.

I do not understand why an assessment of the iteration error would have to await real mission data. I tend to disagree: This is precisely the advantage of applying the technique to a model for which you know the exact solution: It allows to evaluate all the error contributions. By applying the method using synthetic data with zero measurement error and zero random error and a very fine discretization, the limit of the iteration should differ from the exact solution only by the truncation error. For a very fine discretization error, however, this truncation error should be small. One can easily

compare any iteration step to the limit and evaluate how the iteration error decreases. It would similarly be not difficult to examine the combined effect of discretization and truncation: use zero measurement and random error and compare to the solution obtained with the exact solution. And one could similarly evaluate the effects of measurement errors and random errors. I admit that it may be difficult to disentangle the truncation and discretization errors, except if one constructs a model problem that has only linear and quadratic variations, so that the truncation error is known to be zero. In short: I find it a pity that not all the error types are studied (numerically) in this manuscript, as the examples are so inviting and offer the perfect occasion to do so.

Reply:

We agree with the referee on these valuable comments. To address your comments, we have performed detailed analyses and added additional discussion as the following, using the dipole field case as an example.

In Appendix A, we have now evaluated the iteration error. To demonstrate this point, we increase the number of iterations to 1000. We find that the relative errors decrease with the number of iterations as shown in Figure A1. The relative errors indeed converge to minimal values (less than 0.01% for the linear gradients and less than 2% for the quadratic gradients) after $100^{th}$ iteration. To exclude the effect of the truncation error, we hold the configuration of the 7-S/C constellation while scaling down the distances between satellites by a factor of 100. Due to this reduction, the high-order truncation error converges to zero, leaving only the iteration error. Figure A2 shows the relative errors in the absence of the truncation error. We find that the relative errors of the linear gradients decreased to 0 and those of the majority quadratic gradients decreased to less than 0.1%. Therefore, we conclude that the error generated during the iteration process is relatively small given that the number of iterations is above 100.

In Appendix B, we have now evaluated the discretisation error. To introduce the discretisation error, we assume that the magnetic field value at the measurement point is the average along the satellite's trajectory for a duration of 0.25 seconds before and after the point, in the direction of the satellite's motion. We scaled down the distances between satellites by a factor of 100 so that the high-order truncation error converges to zero. Since there is no measurement error, only discretisation error remains. Figure B1 shows the variation in relative errors of the linear and quadratic gradients with respect to the iteration numbers, with a discretisation error introduced. The relative errors converge after $100^{th}$ iteration. The relative errors of linear gradients are less than 0.012%, while those of majority quadratic gradients are less than 0.1%. Therefore, we conclude that the discretisation error is relatively small.

In Appendix C, we have now evaluated the measurement error by introducing 0.1% and 1% measurement error (by a reduction of the measurement magnitude). Other sources of errors are minimized as mentioned above. Figures C1 and C2 shows the variation in relative errors of the linear and quadratic gradients with respect to the iteration numbers for the dipole field case, for 0.1% and 1% measurement errors, respectively. Again, we find that the relative errors converge to minimal values after $100^{th}$ iteration. Indeed, we

find that the measurement error is of the same order as the accuracy of the instrument.

While the div B = 0 test is a useful overall error measure, it would be interesting if the authors could point out what it tells us about each of the types of errors, e.g, how do these errors vary with the discretization step, with the number of iterations, with the level of measurement and random errors?

Reply:

Thanks for the referee's comments. Please see our response above in complementary to Section 4 and Appendix A-C in the revised manuscript.

**Minor comments**

- line 81: environments -> environment

Reply:

The correction has been made accordingly. Thanks.

- line 160: equation -> equations

Reply:

The correction has been made accordingly. Thanks.

- line 165: SPractical -> Practical

Reply:

The correction has been made accordingly. Thanks.

- line 178: better: "⋯ from central differences of the magnetic observation time series"

Reply:

The correction has been made accordingly. Thanks.

- line 201: as the following -> as follows:

Reply:

The correction has been made accordingly. Thanks.

- line 202: in order the solution exists -> in order for the solution to exist

Reply:

The correction has been made accordingly. Thanks.

- line 203: constellation -> constellation

Reply:

The spelling of this word has been modified accordingly. Thanks.

- line 270: The characteristic size of the S/C -> The characteristic size of the S/C constellation

Reply:

The correction has been made accordingly. Thanks.

- line 344: which -> whose – or, even better, just drop "which geometry is demonstrated"

Reply:

The sentence has been dropped accordingly. Thanks.

- line 381: drop "with portion of the number of linear and quadratic gradients" as this is made explicit in the next sentence

Reply:

The sentence has been dropped accordingly. Thanks.

- line 386: so symmetrical model magnetic field -> a symmetric model magnetic field

Reply:

The correction has been made accordingly. Thanks.

- line 387: drop "accurate"

Reply:

The word has been dropped accordingly. Thanks.

- line 387: I am not sure I understand the logic and the meaning of the following two sentences. If I understand it correctly, a better formulation would be "The zero components of the magnetic gradients are calculated with the algorithm and checked. Further evaluation of the algorithm with a less symmetric magnetosphere model could be useful." Please check.

Reply:

The referee understands it correctly. The correction has been made accordingly. Thanks.

- line 392: drop "as the following"

Reply:

The sentence has been dropped accordingly. Thanks.

- line 396: the abbreviation to be used for "seconds" is "s" rather than "sec"

Reply:

The abbreviation has been modified accordingly. Thanks.

- line 398: the discretization errors brought -> the corresponding discretization errors

Reply:

The correction has been made accordingly. Thanks.

- line 453: "fewer": compared to what?

Reply:

This sentence has been rewritten for clarity. Thanks.

- line 457: "to be able to obtained from" -> "to be obtained by" or "to be available from"

Reply:

The correction has been made accordingly. Thanks.

We are very grateful for the referee's valuable comments. We address your suggestions point-by-point below and improve our manuscript presentation throughout. Major changes include the evaluation of discretisation error, iteration error, and measurement error.

In this reply, the comments of the referee are marked in black and the replies in blue. We hope that our revised manuscript is now more accessible to the general readers.

Comments on the revised manuscript entitled

Quadratic Magnetic Gradients from 7- and 9-Spacecraft Constellations

submitted by Chao Shen, Gang Zeng, Rungployphan Kieokaew, and Yufei Zhou

**General comments**

Many concerns have been addressed in the revision. Several specific concerns remain.

Reply:

In this version, we have improved the structure of the manuscript throughout, particularly in the evaluation of discretisation error, iteration error, and measurement error. Please see our replies to your major comments point-by-point below.

**Specific comments**

Abstract and Key Points

- In the review of the original manuscript, key statements were considered too strong. During the revision, only grammatical errors were corrected, so that the statements now read

"Tests for the situations of magnetic flux ropes and dipole magnetic field have verified the validity and accuracy of this approach."

and

"Magnetic flux ropes and dipole magnetic field testing verifies the validity and accuracy of the approach."

Since the essence has not changed, the statements are still too strong. A complete assessment of the accuracy would require studying all error types and the stability of the model inversion procedure, so the term "verified" in combination with "accuracy" is not applicable in this context. The paper is a proof of concept, i.e., a demonstration

of the validity of the approach, and as such the statements may be rephrased as follows:

"The validity of the approach was demonstrated using magnetic flux ropes and dipole magnetic field models."

and

"Magnetic flux ropes and dipole magnetic field testing demonstrated the validity of the approach."

Reply:

The strong statements in the Abstract and Key Points have been rephrased as the referee's advice accordingly. Thanks.

Comparison of new method with analytical modelling:

- Lines 301-305: In the review of the original manuscript (lines 254-258), the reviewer commented "Only total errors after a given number of iterations are discussed. It would be more interesting to get separate assessments of iteration errors and discretisation errors." but this concern was not properly addressed by the authors. Even if "It is not very easy to separate iteration errors and discretisation errors." as stated by the authors in their reply, the limitations of their demonstrations should at least be critically discussed.

Reply:

We agree with the referee on these valuable comments. To address your comments, we have performed detailed analyses and added additional discussion as the following, using the dipole field case as an example.

In Appendix A, we have now evaluated the iteration error. To demonstrate this point, we increase the number of iterations to 1000. We find that the relative errors decrease with the number of iterations as shown in Figure A1. The relative errors indeed converge to minimal values (less than 0.01% for the linear gradients and less than 2% for the quadratic gradients) after $100^{th}$ iteration. To exclude the effect of the truncation error, we hold the configuration of the 7-S/C constellation while scaling down the distances between satellites by a factor of 100. Due to this reduction, the high-order truncation error converges to zero, leaving only the iteration error. Figure A2 shows the relative errors in the absence of the truncation error. We find that the relative errors of the linear gradients decreased to 0 and those of the majority quadratic gradients decreased less than 0.1%. Therefore, we conclude that the error generated during the iteration process is relatively small given that the number of iterations is above 100.

In Appendix B, we have now evaluated the discretisation error. To introduce the discretisation error, we assume that the magnetic field value at the measurement point is the average along the satellite's trajectory for a duration of 0.25 seconds before and after the point, in the direction of the satellite's motion. We scaled down the distances

between satellites by a factor of 100 so that the high-order truncation error converges to zero. Since there is no measurement error, only discretisation error remains. Figure B1 shows the variation in relative errors of the linear and quadratic gradients with respect to the iteration numbers, with a discretisation error introduced. The relative errors converge after $100^{th}$ iteration. The relative errors of linear gradients are less than 0.012%, while those of majority quadratic gradients are less than 0.1%. Therefore, we conclude that the discretisation error is relatively small.

In Appendix C, we have now evaluated the measurement error by introducing 0.1% and 1% measurement error (by a reduction of the measurement magnitude). Other sources of errors are minimized as mentioned above. Figures C1 and C2 shows the variation in relative errors of the linear and quadratic gradients with respect to the iteration numbers for the dipole field case, for 0.1% and 1% measurement errors, respectively. Again, we find that the relative errors converge to minimal values after $100^{th}$ iteration. Indeed, we find that the measurement error is of the same order as the accuracy of the instrument.

- Lines 327/328: In response to the reviewer's comment to the original manuscript (lines 277/278), the author rephrased the statements "The relative error approaches 50%; however, the absolute error is low." to "The relative error approaches 50%; however, the absolute error is just 0.143, which is approaching zero." which creates a new problem because the unit is missing. If measured in "nT.RE^2", the value is 0.143. If measured, e.g., in the SI unit "T.m^2", the absolute error would be about 5800. The quantification of an absolute error in terms of "small" or "large" is meaningful only if compared to a reference value.

Reply:

The unit should be "nT·$R_E$^2", and it has been added.

- Lines 376/377: In the review of the original manuscript (line 321), the reviewer was concerned about the statement "The relative error approaches 50%; however, the absolute error is low." which was not changed during the revision. The authors are asked to adjust the statement along the lines of the previous comment.

Reply:

The statement has been modified, and the unit has been added.

---

## Author Response (AR3)

Dear Editor,

We are very grateful for the referee's valuable comments. In this revised version of the manuscript, we mainly modified the discussion regarding the various types of errors that arise when applying our method in Section 4 and Appendix. We address the referee's suggestions point-by-point below as well as improve the manuscript's presentation throughout.

In this reply, the comments of the referee are marked in black and the replies in blue. We hope that our revised manuscript is now more accessible to the general readers. Thank you so much for your editorial work.

Kind regards,

Chao Shen and Gang Zeng, on behalf of the co-authors

**Replies to the first reviewer**

The authors have responded to the reviewer comments. It is appreciated that also the language has been revised thoroughly.

There are still a few minor points that are unclear, possibly incorrect.

Reply:

We thank the referee for these valuable comments. In this version, we have improved the manuscript throughout, particularly in the evaluation of discretisation error, iteration error, and measurement error as discussed in Section 4 and detailed in Appendix. Please see our replies to your major comments point-by-point below.

Line 20: ropes -> rope

Reply:

The correction has been made accordingly. Thanks.

Line 408: during measurement period -> during the measurement period

Reply:

The correction has been made accordingly. Thanks.

Line 411: accessed -> assessed

Reply:

The correction has been made accordingly. Thanks.

Line 411: spatial resolution can indeed be due to S/C motion during the measurement, but it also can be the S/C motion in between two successive measurements – that

depends on how the instrument works. It would be good to mention that possibility.

Reply:

Another definition of the spatial resolution has been added accordingly in Lines 397 – 398. Thank you.

Line 419: solution -> solutions

Reply:

The correction has been made accordingly. Thanks.

Line 426: "More advanced evaluation ..." : One can study iteration error in the best possible way using artificial examples. I do not see how satellite data would improve your evaluation of this property of the algorithm. I propose to simply drop that phrase.

Reply:

Thanks for the referee's comments. The sentence has been dropped accordingly. Thanks.

Line 431: "the relative errors of the gradients associated to the measurement errors are expected to be the same magnitude as the measurement errors": It is a generally known property of numerical differentiation that the relative errors on the gradients are often much larger than those of the measurements, especially as you are typically differencing values that are very similar in magnitude.

Reply:

This sentence has been dropped accordingly. Thank you.

Additionally, to better evaluate the measurement errors as suggested by your comment below, we impose random Gaussian error to evaluate our algorithm instead of 0.1% decrease for each measurement in Appendix D. Please see our detailed response below.

Line 484-486: "We also note that ... not perfectly synchronized": I appreciate that you mention these two aspects; it would be even better if you could also say why these are important. For instance, if there are systematic measurement errors, that may lead to an error that is not compensated for by this algorithm.

Reply:

Thanks for the referee's comments. Nowadays, the sampling time resolution of the detectors are already very high, and the temporal variations of the magnetic field or magnetic spatial gradients can be obtained from the time-series data. So, this algorithm only focuses on the magnetic spatial gradients, and the magnetic measurements from different spacecraft should be simultaneous. Furthermore, a homogeneous set of instruments onboard the constellation may not be achieved so that systematic error may arise. The total systematic error can be analysed by the well-established error theory. These clarifications have now been incorporated to the manuscript in Section 5 where we provide conclusions and perspectives.

Line 490: drop "expected to be": Can be dropped: you have computed them ...

Reply:

The "expected to be" has been dropped accordingly. Thank you.

Line 495: I think the word "verified" better reflects the results of your work than "demonstrated"; alternatively, you could say "validated"

Reply:

The word "demonstrated" has been replaced by the word "verified" accordingly. Thanks.

Caption figure A1: Make clear that these are relative total errors, just to indicate that this is not the iteration error alone.

Reply:

Thanks for the referee's comments. The phrase "relative errors" has been replaced by "total relative errors" throughout Appendix.

Appendix A: "It can be suggested that the error generated during the iteration process is relatively small." Can you replace this by "The iteration error clearly goes to zero asymptotically." ? If not, I think there is a problem.

Reply:

Thanks for the referee's comments. The sentence has been replaced accordingly.

Figure A2: I would expect to see here plots of the asymptotic values as a function of separation; the variation with iteration number is not relevant here. It is nice to see how the truncation error changes as you reduce the separation. If you decide to stick to just comparing the two separation scales, it would be better to present that comparison in a small table; there is no need for a figure then.

Reply:

Thanks for the referee's comments. The variation of truncation error with the distance between satellites has been investigated in appendix A (see Figure A1).

Idem for Figure B1.

Reply:

Thanks for the referee's comments. The variation of truncation error with the discretization step has been investigated in appendix C.

Appendix C: You state that "each measurement is decreased by 0.1%" That is not a correct way to evaluate the effects of random measurement errors. You should impose gaussian errors with a specified distribution width on the measurements.

Reply:

Thanks for the referee's comments. The random Gaussian error has been imposed to evaluate this algorithm in Appendix D.

Figure C1 and C2: The variation with iteration number is not relevant. You could present the results in the table or make a figure with the error as a function of the magnitude of the measurement error.

Reply:

Thanks for the referee's comments. The random gaussian errors with mean 0 and standard deviation 0.01 are imposed. The following Figure shows the relative total errors of the linear and quadratic gradients at various barycentres for the dipole field case. The measurement error cause large errors for the quadratic magnetic gradients. In Appendix D, it is indicated that the observation data should be filtered to remove noise.